# Cryo-EM structures of plant Augmin reveal coiled-coil assembly, antiparallel dimerization, and NEDD1 binding

Md Ashaduzzaman[1,8] ✉, Aryan Taheri[1,8], Yuh-Ru Julie Lee [2], Yuqi Tang[3], Shubham Mittal [4], Fei Guo[1], Faruck Morcos [4,5,6], Stephen D. Fried [3,7], Bo Liu[2] & Jawdat Al-Bassam [1] ✉

Microtubule (MT) branch nucleation requires Augmin and NEDD1 proteins, which recruit and activate the gamma-tubulin ring complex (γ-TuRC). Augmin is a fork-shaped assembly of eight coiled-coil subunits, while NEDD1 is a β-propeller protein bridging MTs, Augmin, and γ-TuRC. We reconstitute *Arabidopsis thaliana* Augmin assemblies and determine 3.7-7.3-Å cryo-EM structures of its V-junction and extended regions using crosslinking mass spectrometry. These structures reveal a complete plant Augmin model showing multi-coiled-coil interfaces stabilizing its 40-nm hetero-octameric fork architecture. The dual calponin homology (CH) domains at the V-junction terminus adopt open and closed conformations for MT binding. A 12-Å cryo-EM structure shows Augmin undergoes anti-parallel dimerization through conserved surfaces on its extended region. We determine the NEDD1 β-propeller structure with Augmin, revealing direct binding inside the V-junction that enhances dimerization. Direct coupling and evolutionary analyses identify co-varying residue pairs validating the eight-subunit model and NEDD1 interface. Cooperativity between dual CH domains and NEDD1 binding may regulate V-junction binding to MT lattices. This V-shaped dual binding anchors Augmin along MTs, creating platforms for γ-TuRC recruitment and branched MT nucleation.

Microtubule (MT) nucleation is essential for organizing the cytoskeletal networks[1,2]. Across eukaryotes, MT nucleation requires the highly conserved cone-shaped γ-tubulin ring complex (γ-TuRC), which nucleates nascent MTs by templating their tube-like, thirteen protofilament organization[3,4]. The γ-TuRCs nucleate MTs either from MT organizing centers or as branches alongside polymerized MTs[1,2]. In animal cells, the γ-TuRCs localize to centrosomes and nucleate most MTs during interphase, leading to a centralized cellular MT network

with polymerizing dynamic MT plus-ends extending to the cell periphery, while the MT minus ends are anchored to the γ-TuRC-anchored centrosome[1,2]. In contrast, in plant cells, which lack centrosomes, γ-TuRCs often localize along existing MTs and nucleate MT branches to form a near parallel cortical MT network[5,6]. In mitotic cells for both plants and animals, γ-TuRCs are recruited to bind along mitotic spindle MTs and produce parallel MTs extending towards chromosomes in the mitotic spindle[7]. Augmin is required for centrosome-independent γ-

[1]Department of Molecular Cellular Biology, University of California, Davis, CA, USA. [2]Department of Plant Biology, University of California, Davis, CA, USA. [3]Department of Chemistry, Johns Hopkins University, Baltimore, MD, USA. [4]Departments of Bioengineering and Physics, University of Texas at Dallas, Richardson, TX, USA. [5]Department of Biological Sciences, University of Texas at Dallas, Richardson, TX, USA. [6]Center for Systems Biology, University of Texas at Dallas, Richardson, TX, USA. [7]T. C. Jenkins Department of Biophysics, Johns Hopkins University, Baltimore, MD, USA. [8]These authors contributed equally: Md Ashaduzzaman, Aryan Taheri. ✉e-mail: mashaduzzaman@ucdavis.edu; jmalbassam@ucdavis.edu

TuRC-activated MT branch nucleation in mitosis, leading parallel MTs to form bipolar mitotic spindles. The activities of Augmin and Neural precursor cell expressed, developmentally downregulated 1 (NEDD1) are necessary for aligning and segregating chromosomes during cell division, and the defects in its eight subunits lead to long and thin mitotic spindles[7,8]. *Drosophila melanogaster* RNAi screens for mitotic phenotypes identified a subset of the eight Augmin subunits as essential for γ-tubulin association with spindle MTs and producing parallel MT arrays in mitosis[9]. In contrast to the most mitotic functions of Augmin in animals, plant Augmin promotes γ-TuRC MT-branch nucleation in both interphase and mitosis[5–7].

Recombinant γ-TuRCs are weak MT nucleators, and conserved factors recruit, anchor, and activate them at centrosomes or along dynamic MTs[10–12]. Augmin is among the most conserved γ-TuRC-associated factors across plants and animals[13]. The metazoan Augmin complex consists of eight distinct coiled-coil subunits, termed HAUS 1,2,3,4,5,6,7,8[14]. Eight equivalent plant Augmin subunits, named AUG 1,2,3,4,5,6,7,8, were identified in *Arabidopsis thaliana*, revealing their essential function for MT-branch nucleation during both interphase and mitosis[15–17]. Plant genomes, however, include eight distinct AUG8 paralogs, suggesting a diversity of Augmin functions in mediating MT branch nucleation in plant cells[7,16]. In interphase, branched MTs emerge on average at 40° (incident angle) from the polymerized MTs, whereas in mitosis, branched MTs emerge at 10°, leading to a mostly parallel MT array, suggesting a diversity of Augmins with unique mitotic and interphase functions specified by distinct AUG8 subunits[7,16].

In addition to Augmin, other MT associated proteins (MAP) regulate Augmin MT association. TPX2 has been shown to recruit Augmin to MTs in *Xenopus laveis* extracts and is conserved in both plants and animals[18]; however, it is totally dispensable in plants[19]. In contrast, NEDD1, also termed GCP-WD, is a highly conserved MAP across plants and animals, and defects in it critically impacts MT nucleation function in both systems[18,20]. NEDD1 consists of a WD40 β-propeller domain (termed NEDD1-WD β-propeller) and a C-terminal helical coiled-coil[20,21]. Human mutations in NEDD1 or Augmin are linked to neurological disorders and are dysregulated in neural precursor cells[22–24]. Similarly, defects in NEDD1 and Augmin are lethal in plants, suggesting conserved roles and mechanisms in MT nucleation[7,25]. Live cell imaging in *Drosophila* cells during anaphase show that Augmins bind to MTs, followed by γ-TuRC recruitment to nucleate daughter dynamic MTs[26]. A recent study suggested NEDD1 recruits Augmin to bind to MTs and then recruit γ-TuRC to activate MT branch nucleation[20]. Furthermore, these studies suggest that Augmins oligomerize upon binding MTs prior to recruiting γ-TuRC[20]. However, the nature of the oligomerization of Augmin assemblies or their structure remain unknown. The structural mechanisms of Augmin and NEDD1 in regulating γ-TuRC activation along dynamic MTs have also mostly remained poorly understood, in part, due to the lack of reconstitution studies of these systems.

Augmin assemblies are hetero-octamers composed of 30-nm extended region that end with a 10-nm wide V-shaped junction (V-junction), resembling the shape of a "tuning fork". Biochemical studies suggest the Augmin V-junction binds MTs via its conserved dual-headed CH MT binding domain in HAUS 6,7 (AUG 6,7) and positively charged disordered N-terminal region of Haus8 (AUG8), all of which reside at the tip of the long arm of the V-junction[14]. However, it is not clear how the second end of the V-junction stabilizes Augmin binding to MTs. The Augmin dual CH-domains have been compared to the well-studied NDC80/Nuf2 kinetochore complex, which contains a similar dual set of MT binding CH domains[27,28]. The conformation of the dual CH-domains upon MT binding remains unknown. It also remains unknown how Augmins anchor along MTs via their V-junctions and recruit γ-TuRC via their extended region. Recent reports of multiple low-resolution Augmin cryo-EM structures, in combination with AlphaFold 2/ColabFold models, have led to structural models for the

eight coiled-coil assembly, suggesting a general view of the hetero-octameric organization[29–31]. However, even among the clearest of these cryo-EM maps, low-resolution α-helical density is observed, and most of the coiled-coil assembly interactions were inferred by placing AlphaFold2 models into the low-resolution cryo-EM density maps. Difficulties with studying the structures of Augmin structures stem from their highly elongated shape and flexibility, hindering crystallographic or cryo-EM structure determination studies.

Our study presents a comprehensive structural and biochemical analysis of the Augmin assembly. By reconstituting recombinant *A. thaliana* Augmin hetero-octamers and hetero-tetramers assemblies, we utilized crosslinking mass spectrometry (XL-MS) and single particle Cryo-EM to determine structures for different regions of the Augmin assembly, leading to a near-complete de novo Augmin model. Our structural analysis reveals insights into the dynamic flexibility and states of Augmins, such as the conformation of AUG6,7 dual CH-domains at the tip of the V-junction, which adopt both splayed and packed states. Furthermore, we observed that Augmins undergo anti-parallel dimerization, mediated by conserved interfaces along their extended regions. This organization was visualized in a 12-Å cryo-EM structure of the Augmin dimer assembly, highlighting the spatial arrangement and transitions of the extended domain. Our findings also demonstrate that NEDD1 WD β-propeller binding to Augmins requires their V-junction region, and it enhances Augmin dimerization. The 10-Å cryo-EM structure of the Augmin-NEDD1 β-propeller revealed its binding site inside the V-junction. We present direct coupling and evolutionary analyses that reveal pairs of co-varying residues, validating our eight-subunit Augmin model and its NEDD1 β-propeller interface site. Our results lead us to a structural model where Augmin binding to MTs is stabilized by AUG6,7,8 and the NEDD1 β-propeller, positioned on different ends of the V-junction to anchor Augmin along multiple MT-protofilaments. This arrangement likely creates a platform for anchoring γ-TuRC, thereby facilitating MT branch nucleation. Our work significantly advances the understanding of Augmin's role in MT dynamics, providing detailed molecular insights into its interactions and structural organization.

## Results

### Biochemical reconstitution of plant Augmin assemblies

To purify *A. thaliana* Augmin assemblies, coding regions for AUG1,2,3,4,5,6,7,8 subunits were assembled into polycistronic bacterial expression vectors (Supplementary Fig. 1). The AUG1, AUG3, AUG4, and AUG5 subunits consist mostly of highly conserved α-helices, while AUG2, AUG6, AUG7, and AUG8 were relatively less conserved[15]. The MT binding region of AUG8 is highly divergent and predicted to be disordered or unstructured and was thus excluded from expression (Fig. 1A). To overcome many rare Leu and Arg plant codons, codon-optimized AUG2,7,8 sequences were used for bacterial expression. We generated three polycistronic co-expression vectors consisting of either eight subunits (AUG1,2,3,4,5,6,7,8) or four subunits (AUG1,3,4,5 or AUG2,6,7,8) (Supplementary Fig. 1). Polycistronic expression led to soluble assemblies of AUG1,2,3,4,5,6,7,8 and AUG1,3,4,5 (Supplementary Fig. 1). However, AUG2,6,7,8 assembly were unstable and could not be purified, suggesting that AUG2,6,7,8 require the AUG1,3,4,5 for their solubility and can only be studied as a part of the full Augmin (AUG1,2,3,4,5,6,7,8). Mass spectrometry confirms each Augmin subunits is purified in the assembly (Supplementary Fig. 1E). The AUG6 C-terminal region is prone to degradation with multiple polypeptide bands (Supplementary Fig. 1C). Mass spectrometry also identified several low amounts of bacterial contaminating proteins, suggesting that they copurify with Augmin assemblies through multiple steps (see below; Supplementary Fig. 12). We measured the masses of Augmin assemblies using size exclusion chromatography with multi-angle light scattering (SEC-MALS) and mass photometry (MP) showing AUG1,3,4,5 are hetero-tetramers with a

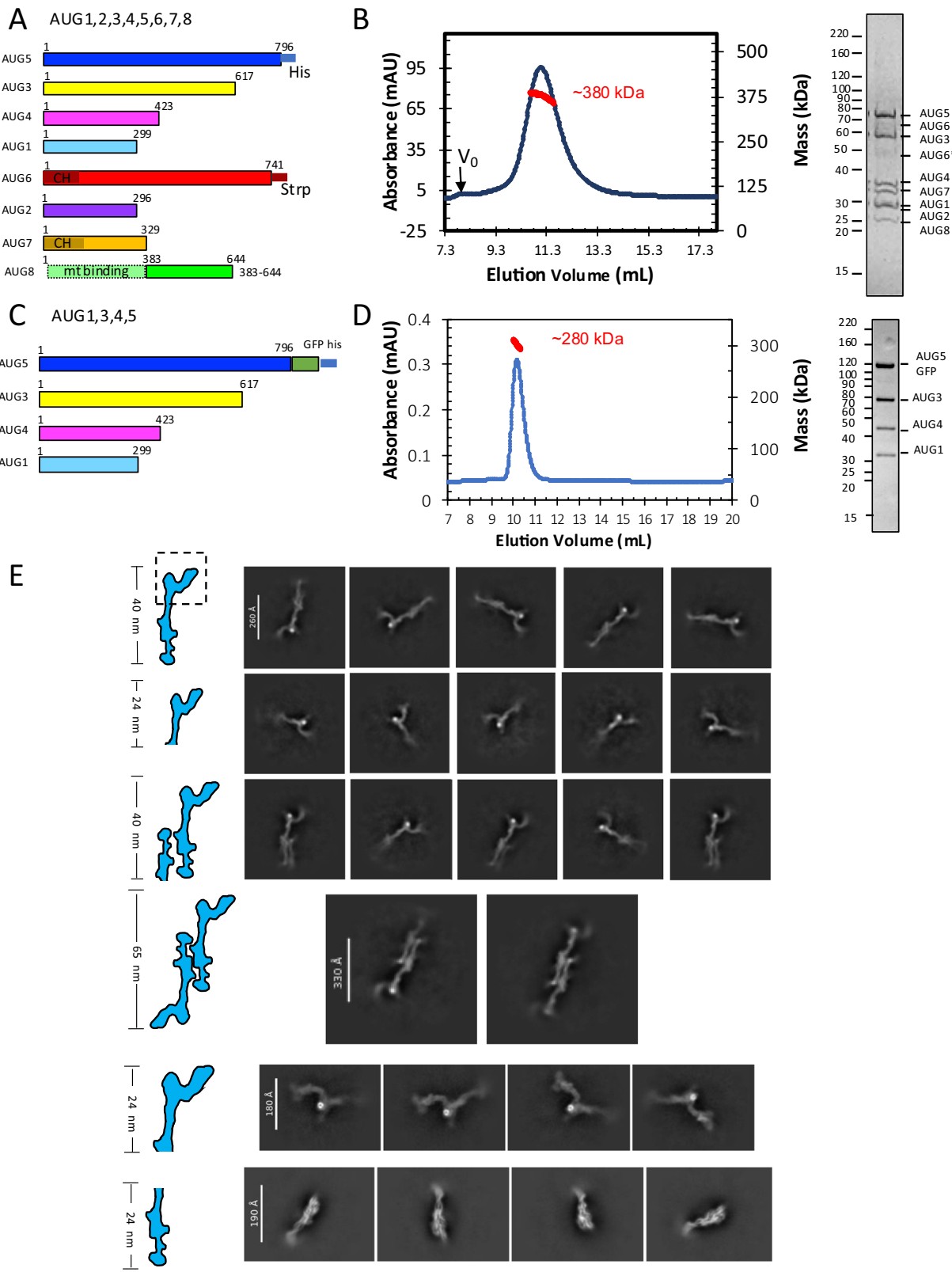

mass of 250–280-kDa mass, and the AUG1,2,3,4,5,6,7,8 assemblies form a hetero-octamers with a mass of ~380–420 kDa mass (Fig. 1B, D and Supplementary Fig. 1D, H). The measured masses suggest a stoichiometry of one Augmin subunit per Augmin assembly, consistent with human Augmin[14]. Although the plant AUG1,2,3,4,5,6,7,8 hetero-octamers are stable, changes in ionic strength led to their destabilization through a cascade of AUG2,6,7,8 dissociation, likely caused by

the degradation or dissociation of AUG6, resulting a mixture of hetero-octamers (AUG1,2,3,4,5,6,7,8) and hetero-tetramers (AUG1,3,4,5) over time (see below Supplementary Fig. 7F).

## Cryo-EM of Augmin reveals its architecture and dimerization

We collected cryo-EM datasets for hetero-octameric (AUG1,2,3,4,5,6,7,8) and Hetero-tetrameric (AUG1,3,4,5) Augmin

**Fig. 1 | Biochemical and structural characterization of *A. thaliana* Augmin assemblies. A** Schematic representation of plant Augmin hetero-octamer (AUG1,2,3,4,5,6,7,8) subunit organization, showing domain boundaries and purification tags. Note: AUG8 construct (residues 383–644) excludes the N-terminal MT binding domain (See Supplementary Fig. 1). **B** Biochemical validation of hetero-octameric Augmin complex. Left: SEC-MALS analysis confirming monodisperse assembly with predicted octameric mass (single run). Right: SDS-PAGE of purified complex showing all eight subunits (See Supplementary Fig. 1). AUG6* marks a degradation product of AUG6. **C** Schematic of minimal Augmin hetero-tetramer (AUG1,3,4,5) subunit organization, including domain boundaries and purification tags. **D** Biochemical validation of hetero-tetrameric complex. Left: SEC-MALS analysis demonstrating monodisperse assembly with tetrameric mass (Single run). Right: SDS-PAGE confirming presence of four subunits. **E** Cryo-EM 2D-class averages revealing Augmin complex architecture. Top panel: Monomeric AUG1,2,3,4,5,6,7,8: 40 nm tuning fork structure. Second panel: Monomeric AUG1,2,3,4,5,6,7,8: V-junction focus with extended region. Third panel: Dimeric AUG1,2,3,4,5,6,7,8: Two extended regions, single V-junction. Fourth panel: Dimeric AUG1,2,3,4,5,6,7,8: Two extended regions, two V-junctions. Fifth panel: Focused V-junction from AUG1,2,3,4,5,6,7,8: 24-nm V-junction and stem regions. Bottom Panel: Monomeric AUG1,3,4,5: 23 nm extended region.

assemblies (Supplementary Fig. 2 and Fig. 1E). 2D-class averages show that the AUG1,2,3,4,5,6,7,8 particles have a characteristic 40-nm tuning-fork shape, similar to previously determined structures of Augmin (Fig. 1E, top panel)[29–31]. 2D-class averages of AUG1,2,3,4,5,6,7,8 particles reveal a 30-nm extended region with a wide center, a narrow leg-like extension, and a foot-like density at one end, which closely resembles the shape of the AUG1,3,4,5 particles (see below). The other end forms a V-junction shape, leading to a second arm with a globular end (Fig. 1E, second panel). The V-junction consistently possesses a high electron density spot. Mass spectrometry analysis is suggestive that this interactor is RplB—a bacterial β-barrel ribosomal protein that crosslinks to AUG5 in this region and may remain associated with rRNA which could potentially explain the high electron intensity of this spot (see below Supplementary Fig. 12 and Fig. 1E). In many cases, the 2D-class averages show the crescent-shaped V-junction without the 24-nm extended region, suggesting this region may be either broken off or out of focus in those images (Fig. 1E, second panel). 2D-classes of 40-nm particles reveal dual extended region densities with a single V-junction (Fig. 1E, third panel). Finally, 2D-classes of 65-nm particles display dimeric, C2-symmetric Augmin particles with dual anti-parallel tuning forks, indicating that Augmin dimers form in an anti-parallel manner via their respective extended regions (Fig. 1E, fourth panel). These diverse types of 2D-class averages suggest that these AUG1,2,3,4,5,6,7,8 particles have a propensity for dimerization, with some particles potentially losing the V-junction, which aligns with the biochemical interpretation of Augmin dissociating into sub-complexes. However, nearly half of the AUG1,2,3,4,5,6,7,8 particles are missing the extended region (Fig. 1E, second panel).

The 2D-class averages for the AUG1,3,4,5 assemblies show ~24-nm elongated filament-like particles with clear secondary structural elements. These particles have a wide profile near their center, a narrow leg-like density connected to a foot-like domain at one end, and an elongated short extension at the other end (Fig. 1E, bottom panel). The 2D-classes of AUG1,3,4,5 particles exhibit similar features to the lower extended region of the AUG1,2,3,4,5,6,7,8 particles (compare Fig. 1E, top and bottom panels). These particles are visible from multiple angles, including end-on views (Fig. 1E, bottom panel; Supplementary Fig. 2). These observations suggest that the foot region is mobile and flexible in the cryo-EM images, despite the more ordered structure of the extended region.

**Single particle cryo-EM structures of the full Augmin assembly**
2D-class averages of the hetero-octameric Augmin particles (AUG1,2,3,4,5,6,7,8) show a mixture of mostly side and top views (Fig. 1E, panels I and II). As a result of the low signal to noise due to the large box sizes and extensive flexibility, we were only able to determine a consensus 9-Å Augmin structure (Supplementary Fig. 2). This reconstruction shows characteristic features of previously seen for the Augmin (Supplementary Fig. 2, lower left)[29–31]. Heterogeneity analysis allowed us to determine a low-resolution reconstruction for the Augmin particles with a second extended section, termed Augmin 1.5, revealing an incomplete Augmin dimer particle (Supplementary Fig. 2, lower middle). We also isolated the Augmin dimer particles and

applied C2 symmetry, leading to a 12-Å resolution reconstruction of the central core dimer interface, termed the Augmin dimer (Supplementary Fig. 2, lower right; see below).

To determine the single particle structures of the Augmin's V-junction-stem region with greater clarity, we combined all particles with the Augmin V-junction by recentering and re-extracting this region from the AUG1,2,3,4,5,6,7,8 datasets, and processing them as described in the scheme presented in Supplementary Fig. 3. The flexibility analyses revealed variable rotation of up to 20° in the long arm of the Augmin V-junction (Supplementary Figs. 3 and 7A–C). We determined two reconstructions for this V-junction-stem region: (1) a 7.3-Å resolution reconstruction representing a state in which the dimeric CH domains are tightly packed, which we termed the "closed state" (Fig. 2A, B, right and middle panel; Supplementary Fig. 3, lower left). (2) a 10-Å resolution reconstruction representing a state in which the CH domains are splayed apart, which we termed the "open state" (Fig. 2A, right panel). In the open state, the bow density rotates downward by ~15° with respect to the base of the V-junction. Vectorial comparisons of the residue-to-residue movements for models for the two states show that AUG2,6,7,8 moves upward, while the AUG3,5 in the base moves upward (Supplementary Fig. 7C). The 7.3-Å cryo-EM map of the closed state shows clear helical connectivity in the base of the V-junction, and that the closed dual-pronged packed CH domains are visible at the end of the V-junction (Fig. 2B, left and middle panel; Supplementary Fig. 5D–I). In contrast, the 10-Å cryo-EM map of the open state shows the rotation of the V-junction and the opening of the dual-pronged CH-domains (Fig. 2B, right panel; Supplementary Fig. 6; see below). These structures reveal the transitions of the V-junction and the globular head domain of AUG6,7 suggesting that their dual CH-domains undergo an open and close transition (Supplementary Fig. 6H–J).

We determined a 3.7-Å cryo-EM structure of the hetero-tetrameric Augmin (AUG1,3,4,5) using the scheme presented in Supplementary Fig. 4. Our biochemical reconstitution of this minimal Augmin assembly led to a high resolution cryo-EM structure that reveals clear helical density and side chain interactions, previously not observed in the metazoan Augmin structures (Fig. 2C and Supplementary Fig. 5J–L)[29–31]. The maps reveal the detailed folding organization of AUG1,3,4,5 subunits in the extended region of Augmin. The end of the AUG1,3,4,5 particle was flexible, and we thus utilized a combination of 3D-variability, flexibility, and local refinement to obtain a 6-Å reconstruction for this region (Supplementary Figs. 4 and 5M). The previous Augmin structures have not resolved the terminal extended region, likely due to its flexibility, coupled with its consistently inaccurate prediction by AlphaFold2 models[29–31]. The helical features of this tripod-shaped terminal domain are clear, allowing for the accurate placement of and morphing of AlphaFold models (Fig. 2D and Supplementary Fig. 5B, C, J–M).

Using the overlap between the cryo-EM density maps of the 7.3–3.7 Å resolution of the V-junction stem and the extended region and their fit into the consensus 10-Å Augmin particle cryo-EM structure, we are able to fully map helical densities across the 40-nm Augmin hetero-octamer particle (Fig. 2B, D, E, Supplementary Fig. 5A, B,

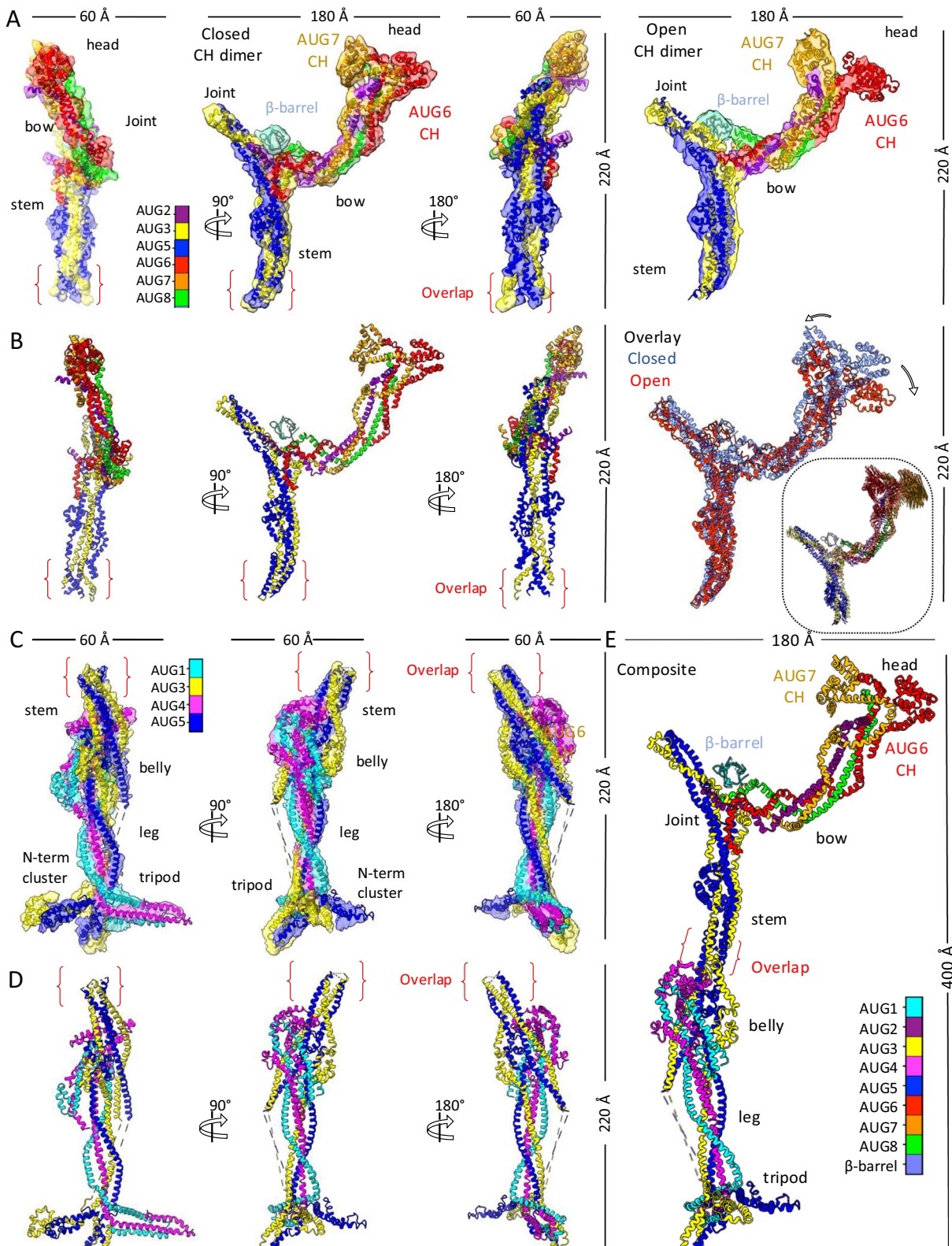

**Fig. 2 | Single particle Cryo-EM structures and models of Augmin assemblies.**
**A** Cryo-EM reconstructions of the AUG1,2,3,4,5,6,7,8 V-junction and stem. Left: 7.3-Å structure with segmented model of AUG1,2,3,4,5,6,7,8 (closed-state AUG6,7 CH-domain dimer). Right: 10 Å structure with open-state AUG6,7 CH-domain dimer (See Supplementary Figs. 2–4). **B** Model analysis. Left: AUG1,2,3,4,5,6,7,8 subunit organization in V-junction and stem. Right: Conformational comparison of closed (red) and open (blue) states. Inset: Vector representation of conformational transition. **C** 3.7 Å reconstruction of AUG1,3,4,5, extended region with segmented model (See Supplementary Figs. 2–5). **D** Detailed subunit organization of AUG1,3,4,5 extended region model. **E** Composite model of complete AUG1,2,3,4,5,6,7,8 complex derived from combined maps and models (See Supplementary Fig. 5A–C).

and Supplementary Movie 1). A globular density, representing a bright spot residing inside the V-junction, potentially corresponds to the ribosomal protein, RplB. Mass spectrometry on the Augmin pulldowns shows RplB to be an abundant contaminant with 518 peptide-spectrum matches, and moreover RplB crosslinks to AUG5 (see below; Supplementary Fig. 12). A β-barrel domain was placed in this density but was excluded from our model building and interpretation (Fig. 1E, fifth panel; Supplementary Figs. 5B, G). The RplB β-barrel proteins likely occupy a "sticky" binding site in Augmin, substituting for a missing binding partner in the bacterial expression system (see below). Hence, our single particle cryo-EM structures lead to full density maps of all structured regions of the Augmin particle, allowing us to build a near-complete structural model for the Augmin hetero-octamer (Fig. 2E and Supplementary Movie 1).

## Complete de novo model of Augmin reveals intertwined coiled-coils

Using our overlapping cryo-EM maps for the Augmin V-junction-stem and the extended regional, we were able to generate a de novo model of the Augmin hetero-octamer (Fig. 2E, Supplementary Fig. 5B, C, and Supplementary Movie 1). The extended region is a 30 nm long assembly with a wide multi-helical region in its center, termed the "belly", connected to a narrow four-helical bundle below, termed the "leg", which terminates into three-prong multi-helical density, termed the "tripod" (Fig. 2E and Supplementary Movie 1). On the other end of the belly region, a four-helical bundle curved structure, we termed the "stem", extends towards the V-junction leading to a nexus point, we term the "joint" (Fig. 2E and Supplementary Movie 1). A short extension in line with the stem extends above the joint forming one arm of the "V". The other arm of the V is composed of a long four helical bundle AUG2,6,7,8, termed the "bow", which ends with a globular region, termed the "head", composed of the AUG6,7 CH domain dimer (Fig. 2E and Supplementary Movie 1).

Our Augmin model explains the multi-subunit coiled-coil helical interactions that stabilize the hetero-octamer and allow for flexibility at its terminal regions (Supplementary Fig. 7). The 30-nm extended region is composed of the hetero-tetrameric AUG1,3,4,5 assembly, while the V-junction is composed of AUG2,6,7,8 C-terminal regions assembled onto a platform composed of AUG3,5 folding back onto themselves (Fig. 3A, B and Supplementary Figs. 5, 7, and 8). The belly region contains AUG1,4, which binds the N- and C-terminal parts the central section of AUG3,5 (Fig. 3B). The belly region is composed of an α-helical bundle (Fig. 3B). The AUG1,3,4,5 C-termini form four helical bundles that supertwist together in the leg region and then fold their C-termini together in the tripod region (Figs. 2D and 3B). In the tripod, the AUG1,3,4,5 diverge into two subdomains where AUG1,4 fold into a larger spoke while the AUG3,5 C-termini form a second shorter spoke (Figs. 2D and 3B). The final and wider spoke in the tripod consists of the AUG3,5 N-terminal bundle, which extends from AUG3,5 N-terminal helices (Fig. 3B and Supplementary Fig. 5M). The tripod region was not predicted accurately by AlphaFold (Supplementary Fig. 8C). From the top end of belly region, the stem of the V-junction emerges from two sets of helices of the AUG3 and AUG5 with opposite topologies (Figs. 2E and 3B and Supplementary Fig. 8C). This region is a platform where AUG3,5 subunits fold back and assemble with AUG2,6,7,8 subunits to form the V-junction (Fig. 3B, top panels). In the AUG1,3,4,5 assembly, the stem is not ordered, likely due to the destabilization of the AUG3,5 foldback zone in the absence of AUG2,6,7,8 (Supplementary Figs. 7F and 8C). The AUG2,6,7,8 C-termini form four helical bundles that can be followed into the V-junction arm, including the AUG6,7 N-terminal dual CH domains (Fig. 3B). Our model differs from previous AlphaFold models in that the helical regions of AUG3,5 foldback zone undergoes more extensive folding with AUG6 and AUG2 C-termini (Figs. 2A, B and 3B and Supplementary Fig. 8A, B).

## Augmin structure stabilized by long and short range coiled-coil interfaces

The structures reveal the full complex topology of all eight Augmin subunits (Fig. 3B). AUG3 and AUG5 form a heterodimeric parallel long coiled-coil that form the backbone of the Augmin structure (Fig. 3B, left panel). AUG1,4 subunits fold into the lower half of the extended region, and they stabilize their N and C-terminal regions by folding in parallel with AUG3,5 N-term cluster in the leg while anti-parallel folding into against themselves into the belly region with both N and C-terminal central helices of AUG3,5, giving this region its girth with an eight helical bundle (Fig. 3B, lower left). At the end of the extended region of Augmin, the tripod consists of a co-folded AUG1,4 C-terminal bundle as the longest spoke and AUG3,5 C-terminal bundle, which forms a second shorter spoke, and the AUG3,5 N-terminal cluster forming the third wider spoke of the tripod (Supplementary Fig. 5M and Fig. 3B, lower panel). In the V-junction of Augmin, the AUG2 and AUG6 C-termini intertwin within the AUG3,5 foldback, forming a six helical bundle leading the stem to widen just below the V-junction (Figs. 2A, B and 3B, top right). The AUG2,6,7,8 C-termini stabilize the AUG3,5 helices in the foldback zone at the top V-junction on while their N-terminal domains towards the long end of the V-junction to form the head globular region (Fig. 3B, top left). The two sets of N- and C-termini of AUG3,5 fold onto AUG1,4 foldback zone, leading to the belly region in the center (Fig. 3B, lower panel). The Augmin model reveals a mixture of parallel and anti-parallel intertwined coiled coils stabilized by short and long-range foldback interactions leading to its conserved organization (Supplementary Fig. 8B, D and Fig. 3B).

We resolved two states of the AUG6,7 CH-domains in the head region: an open and closed state of the AUG6,7 CH domains (Fig. 2B, left panel; and Supplementary Fig. 6G–J). In the closed state, the CH domains are aligned laterally tightly against each other (Fig. 2A, left panel and Supplementary Fig. 6H), and in the open state, they are splayed apart (Fig. 2A, right panel and Supplementary Fig. 6G). These transitions are coupled with a lateral rotation of the AUG2,6,7,8 helical bundle (Fig. 2B, right panel). Taken together, the Augmin V-junction undergoes transitions continuously stretched throughout the structure, where stabilization in one part would likely stabilize subsequent parts (Supplementary Fig. 7). Due to flexibility in this region, we were unable to determine a high-resolution density map of the V-junction (Supplementary Fig. 7).

## Four critical conserved surfaces can be identified on the Augmin structure

To understand the sequence conservation and charge distribution, we aligned sequences for each of the hetero-octameric subunits (AUG1,2,3,4,5,6,7,8) across two hundred orthologs from plants, animals, and insects. We plotted sequence conservation and compared those to a charge distribution plot along the surface of the Augmin model (Supplementary Fig. 9 and Fig. 3D, E). We observe extensive conservation in the coiled-coil helical interactions between subunits within various regions of the structure, suggesting conserved long and short-range assembly interactions (Supplementary Fig. 9A–C and Fig. 3E). Conservation on the surface of the Augmin hetero-octamer reveals four conserved regions with distinct charge distributions, likely indicating critical zones of functional Augmin protein interaction zones (Supplementary Fig. 9A–C and Fig. 3D, E). (1) a large surface on the backside of the belly region involving interfaces with AUG3,5 and AUG1,4 N-terminal helical regions; (2) the AUG3,5 C-termini, and the AUG1,4 C-termini in the tripod (Fig. 3D, E); (3) the top of V-junction interfaces composed of the AUG3,5 central foldback onto the AUG2,6,7,8 C-terminal helical bundle (Fig. 3D, E); this region of the V-junction is bound by the density which we assigned to the RplB β-barrel, which likely forms the binding site for NEDD1-β-propeller (**see below**); and 4) the AUG6,7 dual CH domain dimer in the head region, which mediates MT binding (Fig. 3D-E).

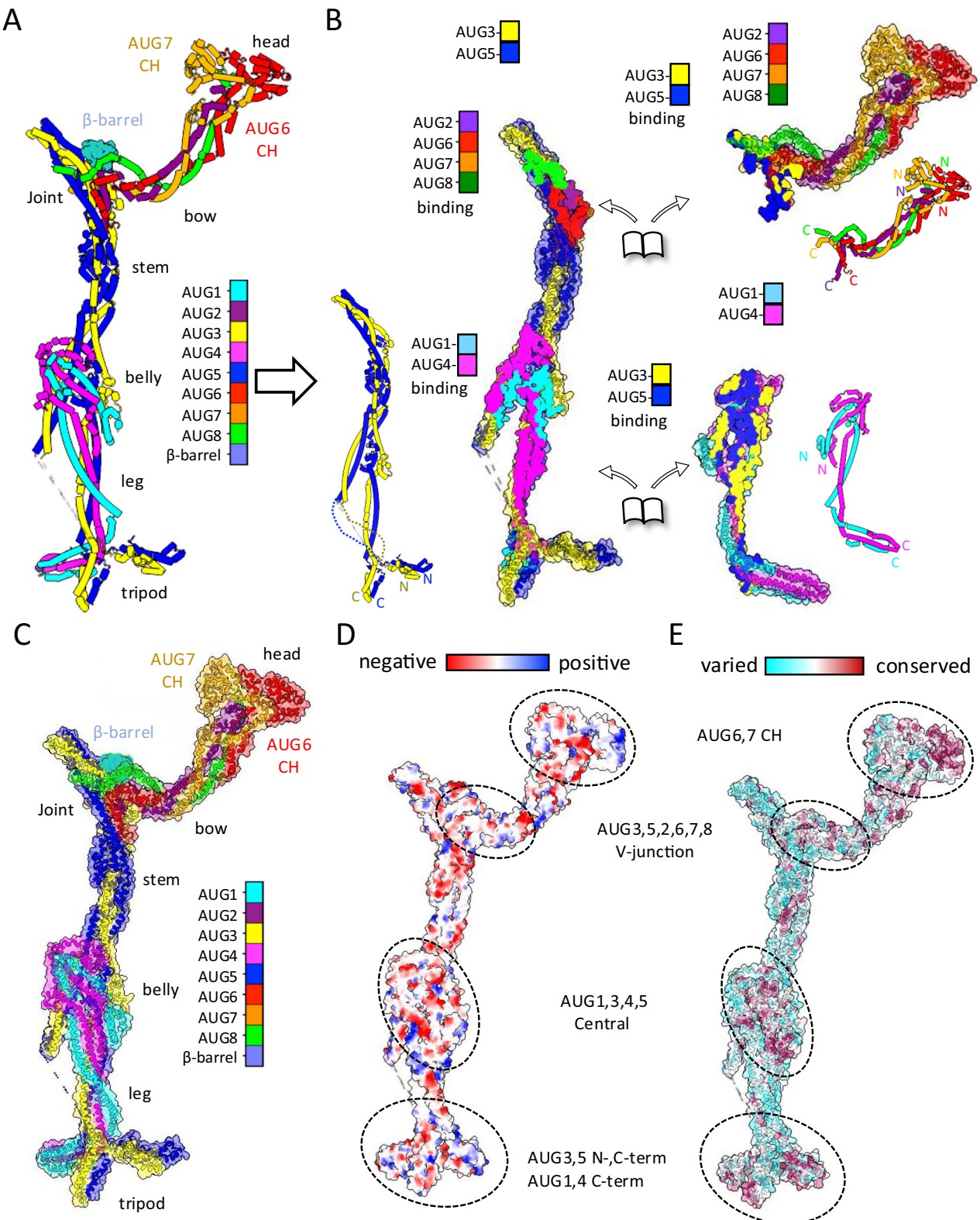

**Fig. 3 | Architectural coiled-coil organization and conservation of the Augmin assembly. A** Ribbon diagram highlighting individual subunit folds and organization. **B** Subcomplex interaction analysis. Left: AUG3,5 heterodimer with interaction footprints. Right: AUG1,4 and AUG2,6,7,8 subcomplexes with corresponding interfaces. Insets: Topological organization of each subunit. **C** Complete Augmin hetero-octamer assembly showing structural domains. **D** Electrostatic surface potential map highlighting functional regions (Supplementary Fig. 9A−C). **E** Surface conservation analysis across plant, animal, and insect (Supplementary Fig. 9F−H).

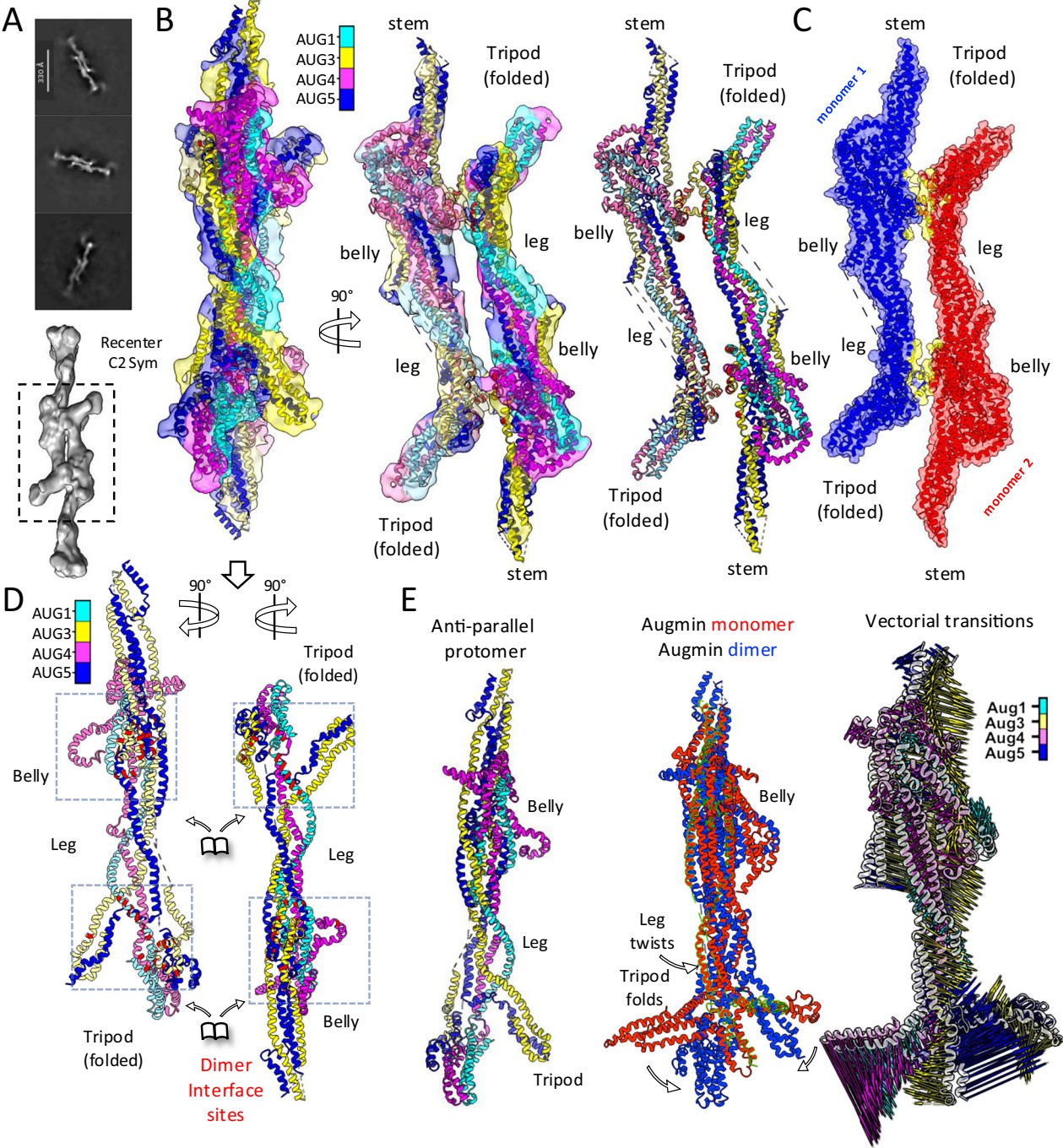

**Fig. 4 | Cryo-EM structure and analysis of Augmin antiparallel dimer assembly.**
**A** Top: 2D class averages showing ~65 nm antiparallel organization of extended domains extracted in 450 pixels box at 1.76 Å/pixel. Bottom: Low-resolution reconstruction with focused refinement of dimerization interface. **B** Detailed structural analysis of extended region dimer. Left: Side view of segmented reconstruction with colored subunit organization. Middle: 90° rotated view showing subunit arrangement. Right: Isolated model highlighting subunit organization at dimer interface. **C** Dimer interface analysis showing protomer organization. Red/ blue: individual protomers. Yellow: Interface region between protomers. **D** Conformational changes in dimer formation. Splayed view of protomers showing AUG1,3,4,5 organization. Interface zones highlighted in red at foot and belly regions. Arrows indicate domain movements during dimerization. **E** Monomer-to-dimer transition analysis. Left: Single protomer structure. Middle: Overlay of monomeric and dimeric states. Right: Vector representation of conformational changes in tripod, belly, and leg regions.

## Augmin undergoes antiparallel dimerization via the tripod to belly regions

2D-class averages of full Augmin hetero-octameric (AUG1,2,3,4,5,6,7,8) particles show side views of a "head-to-tail" Augmin homodimers bound to each other along their long axis (Fig. 1E, fourth panel; Supplementary Fig. 2, left panels). 2D-class averages for the 1.5 Augmin particle show an identical "head-to-tail" dimer of extended regions but are missing a V-junction and stem regions from one of the two assemblies (Fig. 1E, third panel). We determined a 12-Å C2 symmetric anti-parallel cryo-EM single particle structure followed by extended region model placement and de novo model refinement (Supplementary Fig. 2, left panels; Fig. 4). In this structure, the V-junctions and part of the stem are 30 nm away from the center and were plagued with higher flexibility, and were thus excluded (Supplementary Fig. 2, left

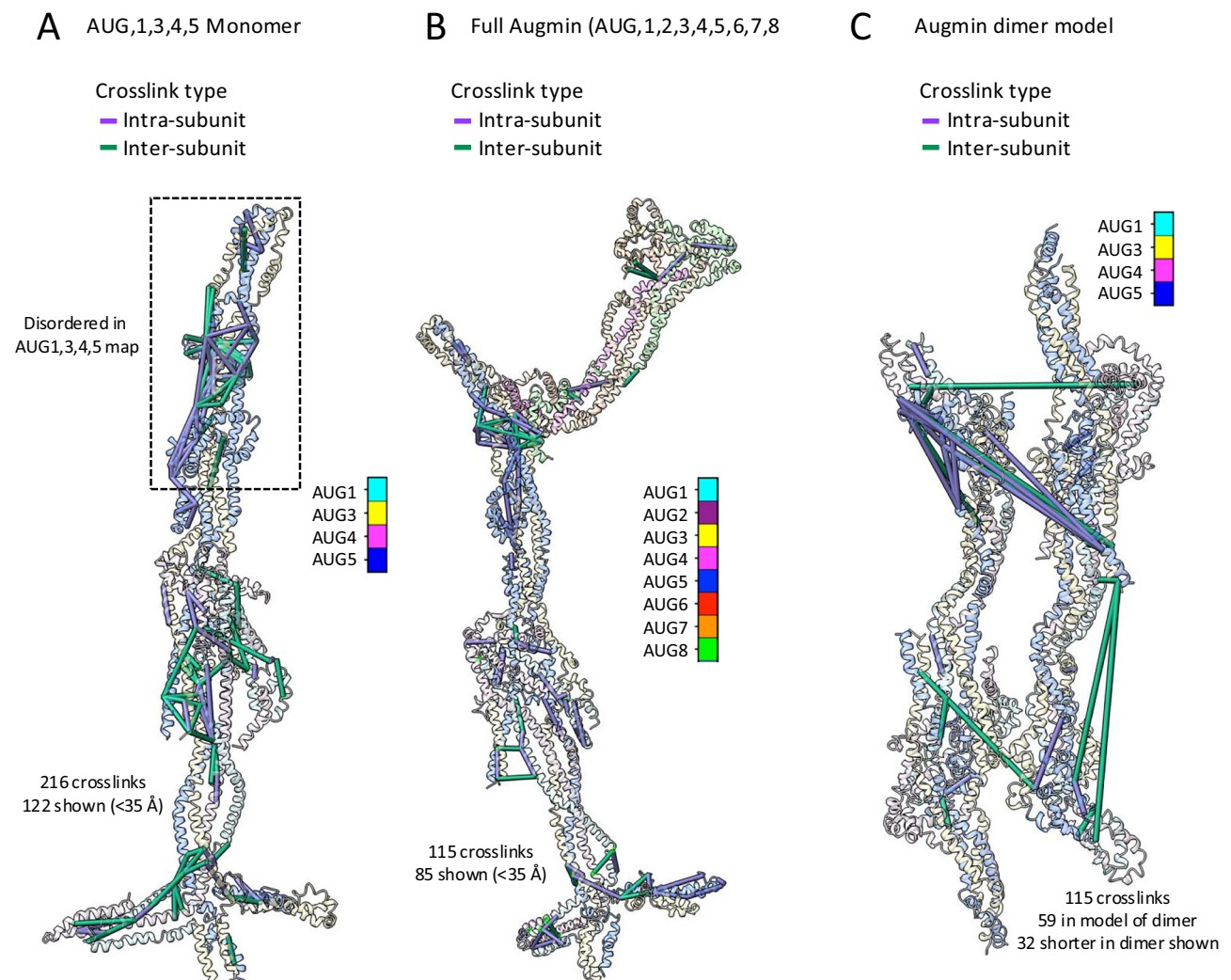

**A** AUG,1,3,4,5 Monomer  **B** Full Augmin (AUG,1,2,3,4,5,6,7,8)  **C** Augmin dimer model

**Fig. 5 | XL-MS validates features of Augmin hetero-tetramer, Augmin hetero-octamer, and antiparallel Augmin dimer. A** Crosslinks identified in AUG1,3,4,5, illustrated in the context of the cryo-EM structural model of AUG1,3,4,5. In total, 216 unique residue pairs were identified, of which the 122 with Cα-Cα distances less than 35 Å are displayed. **B** Crosslinks identified in AUG1,2,3,4,5,6,7,8 illustrated in the context of the cryo-EM structural model of AUG1,2,3,4,5,6,7,8. In total, 115 unique residue pairs were identified, of which the 85 with Cα-Cα distances less than 35 Å are displayed. **C** Crosslinks identified in the AUG1,2,3,4,5,6,7,8 illustrated in the context of the cryo-EM structural model of the anti-parallel Augmin dimer. The 32 crosslinks which have shorter Cα-Cα distances in the context of the anti-parallel dimer compared to the monomer are shown.

panels). Despite the 12 Å resolution of the central region structure, we were able to place the Augmin core models into each Augmin protomer in the head-tail dimer structure and understand the conformational transitions they undertake upon dimerization (Fig. 4B, C and Supplementary Fig. 10). Dimerization is mediated by one face of the belly region and the back side of a rearranged tripod region (Fig. 4B–D and Supplementary Fig. 10). In the Augmin dimer state, the tripod region undergoes a 20° rotational re-arrangement leading repositioning of its three lobes to be face outwards away from the dimerization site (Fig. 4E). The AUG3,5 N-terminal cluster lies close to the dimerization site on the back of the AUG1,4 C-terminal lobe and binds to the belly domain of the second Augmin particle, which is anti-parallelly oriented (Fig. 4D). The belly region and the leg region both twist in their folds, accommodating the rearrangement in the dimeric Augmin state. The surfaces of the tripod and the belly regions mediating the head-to-tail Augmin dimerization are highly conserved in the monomeric Augmin particle (Fig. 3C–E). The Augmin dimer structure reveals the conformational transitions in the extended region during anti-parallel dimerization.

## Crosslinking mass spectrometry validates Augmin assembly interfaces

To understand the multi-subunit coiled-coil interactions within the Augmin hetero-octamers, we carried out crosslinking mass spectrometry (XL-MS) for both hetero-tetrameric (AUG1,3,4,5) and hetero-octameric (AUG1,2,3,4,5,6,7,8) assemblies (Fig. 5; Supplementary Fig. 11I; Supplementary Data, see "Methods"). Two sets of crosslinking analysis were performed, in which top contaminating proteins from bacteria were either excluded or included in the search database (Fig. 5 and Supplementary Figs. 11 and 12). Inclusion of bacterial proteins in the extended XL-MS analysis revealed crosslinks between the Augmin hetero-octamer and several copurifying bacterial proteins. This analysis was used as the basis for identifying RplB protein which binds to the V-junction of AUG1,2,3,4,5,6,7,8 (see above; Fig. 2). Overall, we consider the results from the more restricted search database to be more authoritative, as they yielded more identifications and because the crosslinker used (BS3) is non-cleavable, making it less suited for searches against larger databases (cf. Supplementary Data 1). Therefore, the results presented below are based on the restricted search.

Plotting the Cα-Cα distances for the 216 identified crosslinks that can be fit within the Augmin hetero-tetramer (AUG1,3,4,5) cryo-EM model reveals that 122 (56%) of these crosslinks fall below a 35 Å cut-off, and are shown in Fig. 5A (see also Supplementary Fig. 11A, D, G). When we interrogated where in the structure crosslinks occurred, we found many close-range inter-subunit and intra-subunit crosslinks within the belly region, the foldback region, and the tripod assembly, supporting the assignment of the chains in the cryo-EM density (Fig. 5A). Notably, we detected nine crosslinks that connect residues far in primary sequence, linking the AUG3,5 C-termini to the AUG3,5 N-termini, and 41 crosslinks between AUG3 and AUG5 in the foldback zone (residues 225–378 in AUG3 and residues 342–495 in AUG5) supporting a key feature of the structural model that the AUG3 and AUG5 are elongated coiled-coils that originate in the tripod assembly proceed to fold back, and terminate back in the tripod assembly as well. Crosslinks whose Cα-Cα distances exceed the 35 Å threshold involve the residues assigned to the tripod, suggesting that this region is more dynamic in the hetero-tetrameric (AUG1,3,4,5) assembly. The extensive crosslinks in the AUG3,5 central foldback zone that fall within the 35 Å cutoff are notable, given that this region was not resolved in the AUG1,3,4,5 cryo-EM structure. The data suggest that this region may be too flexible to be resolved by cryo-EM in the absence of AUG2,6,7,8, but the overall structure of AUG3,5 is nevertheless supported by XL-MS (Fig. 2C and Supplementary Figs. 8C and 11D, E).

Plotting the Cα-Cα distances for the 115 identified crosslinks that can be fit within the Augmin hetero-octamer (AUG1,2,3,4,5,6,7,8) cryo-EM model reveals that 85 (74%) of these crosslinks fall below a 35 Å cut-off and are shown on the model in Fig. 5B (see also Supplementary Fig. 11B, E, H). That a higher percent of the crosslinks comports with the cryo-EM model, and suggest the octamer is a less dynamic assembly, which is consistent with our biochemical studies (Fig. 1 and Supplementary Fig. 1). We observed a few inter-subunit and intra-subunit crosslinks in the region forming the V-junction, where the AUG2,6 C-termini interact with the AUG3,5 foldback zone. The AUG2,6,7,8 crosslinks are observed in both the bow region and the AUG6,7 dual CH domains in interfaces with AUG2,8 (Fig. 5B and Supplementary Fig. 11E, H).

We measured the Cα-Cα distances between residue pairs in both the context of the monomeric Augmin assembly and the anti-parallel dimer model. Because the dimer model lacks the V-junction, only 59 of the 115 crosslinks could be mapped to the dimer model, but of those 59, 32 had even shorter Cα-Cα distances in the anti-parallel dimer model, suggesting that inclusion of the anti-parallel dimer state allows us to better capture some of the crosslinking data (Fig. 5C and Supplementary Fig. 11C, F). Of those 32 crosslinks, we identified 8 crosslinks in the tripod and leg regions spanning across dimers (Fig. 5C). The tripod and leg regions undergo conformational changes and become closer in the Augmin anti-parallel dimer model (Figs. 4 and 5C). Thus, these XL-MS data are consistent with our de novo Augmin model and support antiparallel dimerization of the extended region via the conserved interfaces in the belly and tripod regions (Figs. 3E and 5B, C).

### NEDD1 WD-β-propeller binds Augmin via AUG2,6,7,8 subunits

We next reconstituted the NEDD1 WD β-propeller with the hetero-tetrameric (AUG1,3,4,5) and hetero-octameric (AUG1,2,3,4,5,6,7,8) Augmin assemblies to define the location of the NEDD1 Augmin binding site (Fig. 6A, B). Full-length *A. thaliana* NEDD1 was insoluble in bacteria or insect cells. In contrast, the conserved N-terminal NEDD1-WD β-propeller (residues 1–315) is soluble when expressed in insect cells and purified as monodisperse protein (Fig. 6A, B and Supplementary Fig. 13A, B). The NEDD1-WD β-propeller was previously observed to bind MTs and synergize with Augmin[20]. We used sucrose density gradients to explore NEDD1 WD β-propeller binding to AUG1,3,4,5 and AUG1,2,3,4,5,6,7,8 (Fig. 6B and Supplementary Fig. 13C–H). The AUG1,2,3,4,5,6,7,8 bound NEDD1-WD β-propeller, and

they both co-elute at high molecular weight with Augmin subunits, with excess NEDD1-WD eluting at lower molecular weight (Supplementary Fig. 13F–H). In contrast, NEDD1-WD β-propeller did not co-elute with AUG1,3,4,5 (Supplementary Fig. 13C–E). Thus, the NEDD1 β-propeller binding site is missing in the AUG1,3,4,5, but is present in AUG1,2,3,4,5,6,7,8 (Fig. 6B and Supplementary Fig. 13C–H). We validated the AUG1,2,3,4,5,6,7,8 interaction with NEDD1 WD-β-propeller using size exclusion chromatography (Fig. 6C, D). MP on sucrose density purified NEDD1-WD β-propeller AUG1,2,3,4,5,6,7,8 compared AUG1,2,3,4,5,6,7,8 to suggest similar mass (450 vs 460 KDa); however, NEDD1-WD β-propeller binding slightly induces Augmin dimerization (987 kDa), compared to its absence, where Augmins are present mostly as monomers in solution (Supplementary Fig. 13I, J). These data are consistent with location of the NEDD1-WD β-propeller binding site to be only present in AUG1,2,3,4,5,6,7,8, and is missing in AUG1,3,4,5, suggesting it is lies in the V-junction, which is composed of AUG2,6,7,8 (Fig. 6A–D and Supplementary Fig. 13A–F). NEDD1 binding also mildly enhances the dimerization of Augmin particles (Supplementary Fig. 13I, J).

### Cryo-EM structure of Augmin-NEDD1-WD β-propeller reveals its V-junction binding

To determine the NEDD1-WD-β-propeller binding site on Augmin, we collected cryo-EM data for complexes of AUG1,2,3,4,5,6,7,8-NEDD1-WD-β-propeller. Initial 2D class averages show a 50% bigger mass inside the V-junction where we the unknown barrel-shaped density binds Augmin. These 2D classes also shows increased proportions of Augmin dimers and no 1.5 Augmin classes. We then determined a 10-Å cryo-EM structure revealing a larger mass, with circular shape and central hole, attached to the base of the V-junction (Fig. 6E–G). This density is flipped by 90° and has a wider and circular "donut" like shape. Although this map is of lower resolution, we observe conformational change in the V-junction region, tilting upwards, upon placement and morphing of our previously determined Augmin model. We were able to place the AlphaFold predicted NEDD1-WD β-propeller model into the additional density inside the V-junction, revealing that the NEDD1 β-propeller binds via its narrow side interface orientation inside the Augmin V-junction (Fig. 6F, H, I). We plotted the conservation of plant and animal NEDD1 orthologs on the fitted NEDD1 WD-β-propeller AlphaFold model (Fig. 6I). The conservation plot shows that the most conserved surface residues in NEDD1-WD β-propeller overlaps extensively with the interface between the fitted NEDD1 and Augmin V-junction model (Fig. 6I). Furthermore, our low-resolution cryo-EM structure and resulting model for the NEDD1-β-propeller-Augmin V-junction is fully consistent with our biochemical reconstitution studies (Fig. 6B–D). The proximity of the NEDD1-WD β-propeller binding site at the base of the V-junction to AUG6,7 MT binding CH-domain dimer at the tip of the V-junction suggests that NEDD1-WD β-propeller MT binding may stabilize the conformation of Augmin's V-junction to the MT lattice. We propose that the NEDD1-WD β-propeller binds both Augmin and MTs. The Augmin AUG6,7 CH-domain dimer also binds to MTs. The dual binding of Augmin-NEDD1-WD β-propeller likely leads to extensive contact with multiple MT protofilaments (model described below; Fig. 8A).

### Coevolutionary analysis supports Augmin organization and its NEDD1 binding site

We investigated the co-evolution of AUG1,2,3,4,5,6,7,8 subunits and NEDD1 across the eukaryotic kingdoms (Supplementary Fig. 15A) which revealed conservation of all the nine proteins across plants and metazoans, including chordates such as mammals and fish, amphibians, and a small subset of insect species (Fig. 7A and Supplementary Fig. 15A). A significant number of flowering plants were found to possess the complete set of Augmin subunits and NEDD1, highlighting their crucial role in MT nucleation within plants (Fig. 7A, B and

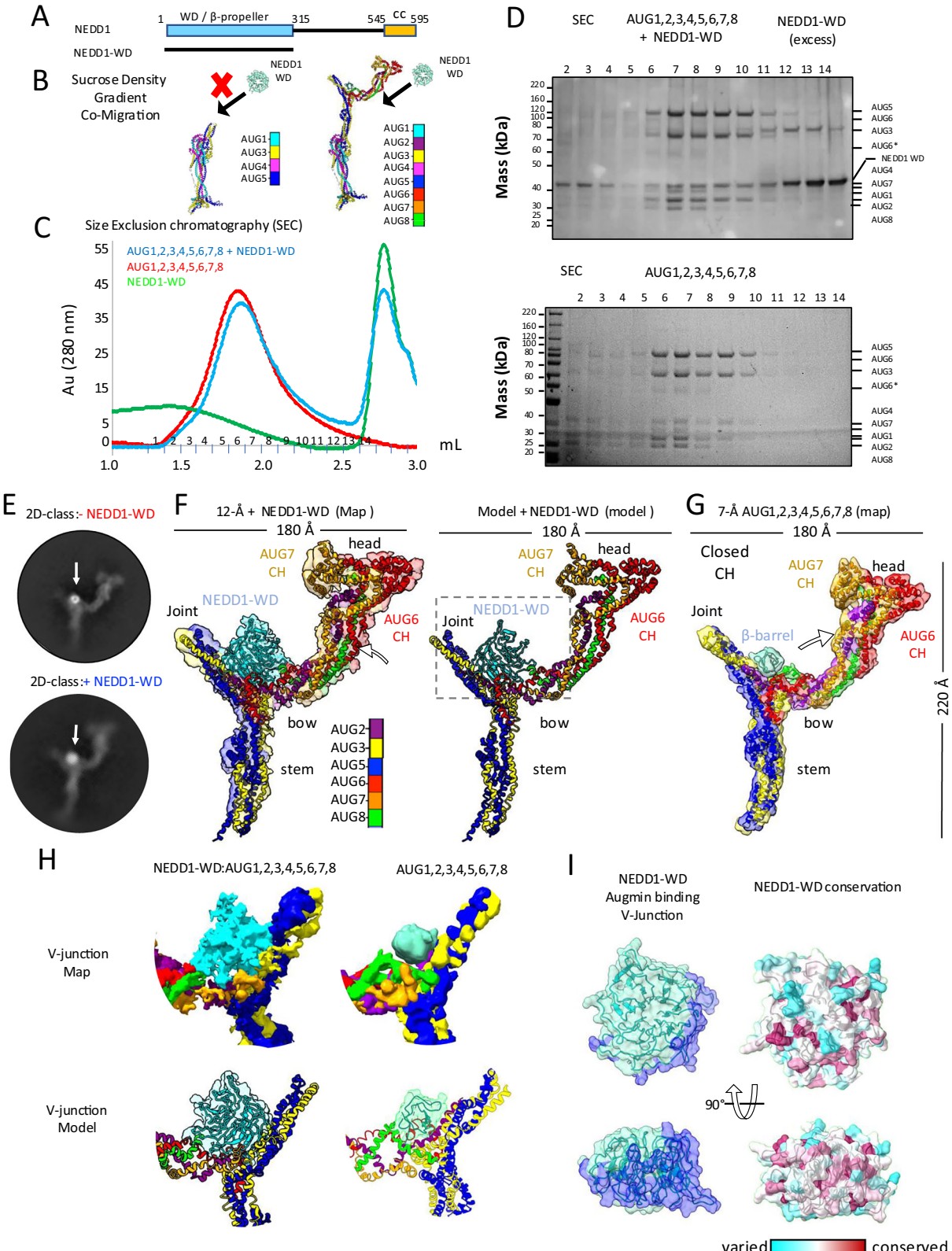

To explore the co-evolution of Augmin hetero-octameric subunits, we designed a workflow that computed coevolutionary couplings of residues obtained from multiple sequence alignments (MSA) of pairwise combinations of Augmin subunits (see "Methods"; Supplementary Fig. 15D). We evaluated twenty pair-wise combinations of the Augmin subunits (Supplementary Table 1) and implemented Direct coupling analysis (DCA) to evaluate each MSA of the subunit pairs. DCA was used to quantify and rank the direct couplings between residue pairs based on Direct Information (DI) hereafter denoted as DI pairs (Supplementary Fig. 15D). DI and related metrics have been shown to be a proxy for physical contacts in protein complexes[32–34] Only statistically significant DI pairs ($p$-value ≤ 0.2) were retained, including 42 DI pairs (Table S2), which we displayed on the cryo-EM Augmin model (Fig. 7C) and

**Fig. 6 | Biochemical reconstitution and structure NEDD1 WD β-propeller-Augmin complex. A** NEDD1 domain organization. N-terminal WD40 β-propeller domain (termed NEDD1-WD β-propeller). C-terminal helical domain (Note: Only WD β-propeller domain successfully purified). **B** Complex formation analysis: AUG1,2,3,4,5,6,7,8 binds NEDD1-WD β-propeller and while AUG1,3,4,5 shows no binding (See Fig. S12C–H). **C** SEC analysis of complex formation. Red: AUG1,2,3,4,5,6,7,8 alone. Blue: AUG1,2,3,4,5,6,7,8 with NEDD1-WD β-propeller. Green: NEDD1-WD alone. **D** Biochemical validation of SEC analysis in (**C**) by SDS-PAGE showing co-migration of NEDD1-WD with AUG1,2,3,4,5,6,7,8 and comparison with AUG1,2,3,4,5,6,7,8 alone (*N* = 2). **E** Cryo-EM 2D-class average comparison NEDD1-WD β-propeller binding to Augmin. Top: AUG1,2,3,4,5,6,7,8 V-junction-stem with β-barrel density. Bottom: AUG1,2,3,4,5,6,7,8-NEDD1-WD β-propeller complex. **F** Structural characterization of complex. Left: 12 Å Cryo-EM segmented reconstruction with fitted models. Right: Ribbon representation of complex. **G** 7.3-Å model showing β-barrel density bound to V-junction stem. Note the marked conformational change. **H** Comparative analysis of V-junction. Top: Segmented maps of complex vs. AUG1,2,3,4,5,6,7,8 alone. Bottom: Atomic models of NEDD1-WD and β-barrel regions. **I** NEDD1-WD-β propeller interface analysis. Right: Surface views of Augmin interaction site. Left: Conservation analysis across species.

display on linear AUG1,2,3,4,5,6,7,8 connectogram (Fig. 7D). Mapping the DI pairs onto the structural model of AUG1,2,3,4,5,6,7,8 reveals strong correspondence in subunit pairings across the Augmin V-junction, dual-CH domains, extended, the tripod regions (Fig. 7C).

We performed a coevolutionary analysis of NEDD1-WD β-propeller with Augmin subunits using the seed sequences from humans, plants and insects (see "Methods", Supplementary Fig. 15C). A species occurrence heatmap was constructed, revealing the co-evolution of NEDD1 and the eight Augmin subunits across species from various taxonomic groups (Fig. 7B). The evolutionary analysis identified 461 species which contain all 9 proteins including 269 flowering plants, 180 chordates and 12 insects (Supplementary Tables 3 and 4 and Supplementary Fig. 15A). To independently identify the NEDD1-WD β-propeller binding site on the Augmin hetero-octameric assembly, DCA was carried out between NEDD1-WD β-propeller and AUG2,6,7,8 subunits (see "Methods"). The identified DI pairs between NEDD1-WD β-propeller and Augmin subunits were then used to guide a molecular dynamic simulation, which brought them together at the V-junction of the Augmin hetero-octamer, revealing a model that is very similar to the cryo-EM structure with NEDD1-WD β-propeller binding to the AUG2,6,7,8 V-junction (Supplementary Table 5 and Figs. 7E and 6). These findings offer an independent line of evidence supporting the biological relevance of the functional interfaces between Augmin hetero-octamer and the NEDD1-WD β-propeller.

## Discussion

We have biochemically reconstituted and determined single particle cryo-EM structures of *A. thaliana* hetero-octameric (AUG1,2,3,4,5,6,7,8) and hetero-tetrameric (AUG1,3,4,5) Augmin assemblies (Figs. 1 and 2 and Supplementary Movie 1). Our ability to reconstitute the multiple types of assemblies, apply single particle recentering, flexibility, and heterogeneous refinement strategies allowed us to achieve medium to high resolution for the majority of the Augmin structure, not achieved in past studies[29–31]. Due to the improved resolution in these Augmin structures, we were able to produce a composite cryo-EM structure from our different maps of Augmin that allowed us to produce a de novo model of the plant Augmin hetero-octamer (Supplementary Movie 1). This plant Augmin hetero-octamer model is supported by extensive XLMS studies that are consistent with the structural placement of the subunits and interactions among them (Fig. 5). Our plant Augmin model differs extensively from the metazoan AlphaFold or ColabFold derived models placed into the lower-resolution cryo-EM structures (Supplementary Fig. 16)[29–31]. We also resolved two states, open and closed, for the AUG6,7 CH-domain dimer in the head of the Augmin V-junction region associated with changes in the V-junction arm (Fig. 2 and Supplementary Movie 1).

### Structural roles of Augmin regions in dimerization and NEDD1 binding

Our biochemical and structural studies uncovered Augmin's propensity for dimerization (Fig. 4). Biochemical reconstitution of NEDD1 WD β-propeller with Augmin reveal its binding site inside the Augmin V-junction and its ability to enhances Augmin dimerization. Our cryo-EM data reveals that Augmin particles form anti-parallel dimers mediated via two binding sites on the extended section in the belly and tripod regions. We show the two dimerization interfaces are highly conserved across Augmin orthologs in plants and animals (Supplementary Fig. 9 and Fig. 3E). Our cryo-EM structure of the Augmin-NEDD1-WD β-propeller reveal the NEDD1 binding site resides along the AUG2,6,7,8 inside the V-junction, but likely induces conformational changes in the Augmin extended region, promoting dimerization. These structures and biochemical studies a provide a revised view of the long-distance cooperativity within Augmin between its NEDD1/MT binding sites at V-junction and its γ-TuRC binding site, likely residing along its extended region. The V-junction binding to NEDD1 WD β-propeller and MTs likely promotes the extended region dimerization, generating a platform for γ-TuRC binding (model described below).

### The coevolution of Augmin hetero-octamer and NEDD1 across eukaryotes

We evaluated the conservation and co-evolution of hetero-octameric Augmin subunits as an assembly and their interaction with NEDD1, revealing the conservation of all nine proteins across plants and metazoans, but Augmin and NEDD1 were not conserved across fungi, protozoa, suggesting that either NEDD1 or Augmin subunits are either absent or has undergone extensive evolutionary divergence (Fig. 7A). The coevolutionary map revealed collective crucial and conserved functional roles of Augmins and NEDD1 in regulating MT nucleation (Fig. 7C–E). DCA analyses revealed 42 DI pairs across the hetero-octameric Augmin assembly interactions, independently validating their role in forming the Augmin structure (Fig. 7C, D). DI pairings from DCA also validate the NEDD1-WD Augmin V-junction interface, suggesting the critical role of NEDD1 WD β-propeller in facilitating Augmin function (Fig. 7E).

### A two-site MT binding model for Augmin V-junction and NEDD1

Our Augmin structural and biochemical studies allow us to develop an Augmin MT binding site relation to NEDD1-WD β-propeller, and its structural organization as shown in Fig. 8. Recent total internal reflection fluorescence reconstitution studies of human Augmin and γ-TuRC in MT branch nucleation by Zhang et al. dissected the relationship of between NEDD1 and Augmin in binding the MT lattice[20]. The initial binding of Augmin to MTs is dynamic and diffusive. However, Augmin signal increases by two to threefold, leading to static Augmin binding along MTs, suggesting Augmin oligomerization may impact high-affinity MT binding[20]. After this transition, Augmin(s) efficiently recruit γ-TuRC to activate branch MT nucleation. Augmin binding to MTs is stabilized by NEDD1-WD β-propeller binding to MT lattices. Our Augmin structures provide a potential explanation for this process. We compared our Augmin structures with conformation of AUG6,7 CH domain dimer in open and closed states and Augmin-NEDD1 WD β-propeller state, all modeled from our structures, by docking them onto the MT lattice, as shown in Fig. 8A. We overlaid the Augmin via the dual AUG6,7 Ch-domain onto the NDC80/Nuf2 kinetochore dual CH domain dimer in their MT bound states (Fig. 8A, right)[27]. In our docking, the Augmin's V-junction lays along MTs bridging across multiple protofilaments (Fig. 8A). Due to the conformational changes in the V-junction of each of the states of Augmin (Supplementary Movie 1),

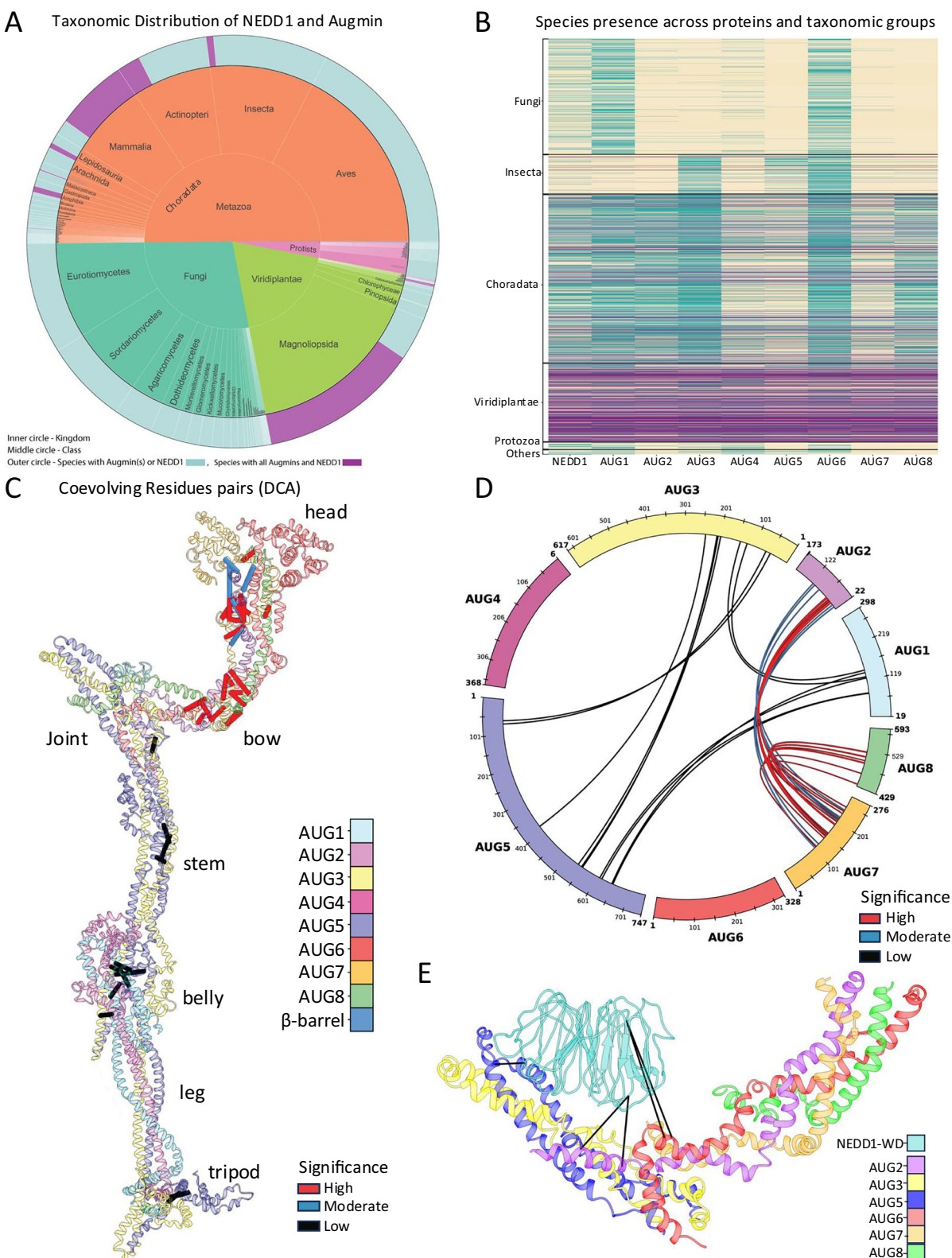

the footprint of each of the V-junction regions of closed, open, and NEDD1-WD β-propeller bound states are compared. Comparison of the size of MT footprint of each of the Augmin v-junction in of the three states on interaction with the MT binding sites suggests that Augmin AUG6,7 CH-domains dimer in the closed state likely binds with a higher affinity along MTs compared to the open state (Fig. 8A, middle and right panels). The binding of NEDD1-WD β-propeller to the V-junction forms a second crucial MT binding site that likely stabilizes Augmin

V-junction interaction with multiple MT protofilaments (Fig. 8A, left panel). With this stabilized dual binding interface, NEDD1-WD β-propeller bound Augmin becomes more tightly bound. We believe this organization may explain the "force-bearing" properties of Augmin in stabilizing MT branch nucleation sites. This binding arrangement rationalizes the V-junction shape of the interface, which leads the 30-nm Augmin extended region to become propped above the MT surface (Fig. 8A and Supplementary Movie 1).

**Fig. 7 | Coevolutionary analysis and direct coupling analyses support the Augmin hetero-octameric assembly structure and its interface with NEDD1-WD β propeller. A** Circular summary of eukaryotes with Augmin subunit(s) and NEDD1. Concentric circles encode taxonomy (center−kingdom; middle−class. The outer rim annotates conservation of the eight Augmin subunits and NEDD1 across species (pink−conservation; light cyan−absence of one or more subunits). A complete evolutionary tree is shown in Supplementary Fig. 15A. **B** Line-based co-evolutionary representation of Augmin subunits and NEDD1 across eukaryotes. Each line represents a unique species. Line color code: yellow−protein is absent; green−protein is present; purple−complete set (all eight Augmin subunits plus NEDD1 in

that species). **C** Structural model of the Augmin hetero-octamer displaying DCA pairs as bars across Augmin subunits. Pairs are colored by significance (red: $p \leq 0.05$, blue: $0.05 < p \leq 0.1$; black: $0.1 < p \leq 0.2$). The $p$-values were calculated using right tailed Hypergeometric test, testing for enrichment of true positive inter-domain DCA contacts among a given number of predicted DI pairs. Full pair list is in Supplementary Table 3. **D** Circular view of the sequences in Augmin subunits with the DCA pairs colored as in (**C**). **E** Augmin hetero-octamer V-junction model displaying the NEDD1-WD β-propeller and its DCA pairing (black bars) to AUG2,5,6,8, as detailed in Supplementary Table 5.

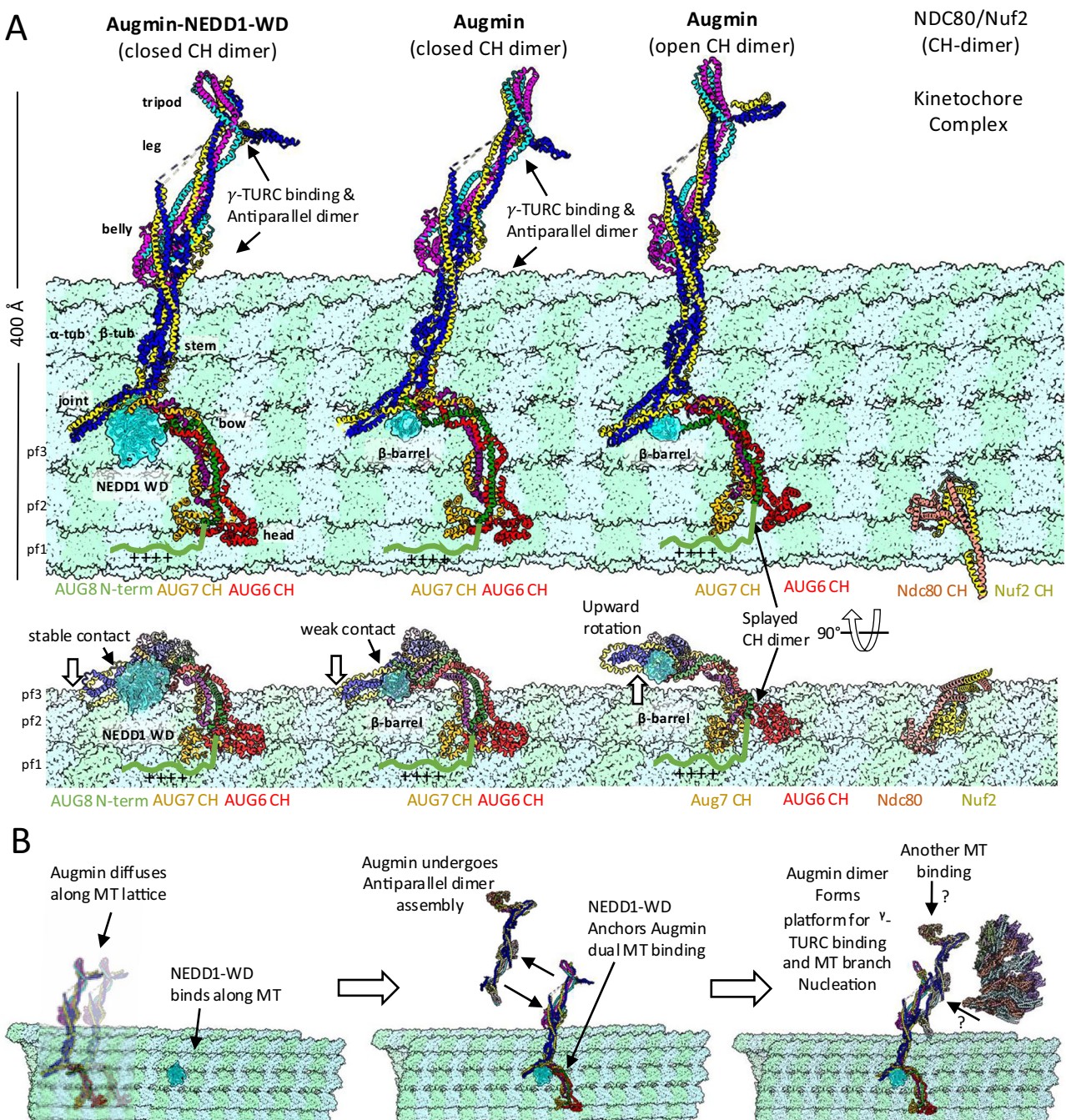

**Fig. 8 | Mechanistic model of Augmin-NEDD1-MT interactions and MT branch nucleation. A** MT lattice binding modes. Left: NEDD1-WD complex with closed AUG6,7 CH-domains. Center-left: β-barrel density impact on MT binding. Center-right: Open CH-domain configuration. Right: NDC80/Nuf2 CH-domain reference structure views shown in a side and a 90° rotated orientation. **B** Proposed mechanism for Augmin function. Left: Transition from diffuse to NEDD1-WD-bound state. Middle: Formation of anchored complex and dimerization. Right: γ-TuRC recruitment and MT branch nucleation.

## Augmin dimers may anchor the γ-TuRC along MTs in branch nucleation

Our MT binding model presents several testable hypotheses regarding the roles of Augmin, NEDD1, and γ-TuRC domains in assembling the MT branch junction machinery (Fig. 8B). Augmin's extended regions likely dimerize, as visualized in our structure (Fig. 4), creating larger interfaces to bind and activate γ-TuRC complexes, anchoring them more efficiently along MTs to promote MT branch nucleation. NEDD1's helical coiled-coil C-terminal domain may facilitate Augmin dimerization by forming tetrameric oligomers, which may γ-TuRC binding by interacting with Mozart and the GCP6 N-terminal region, as described in a recent structural study[35]. Augmin oligomeric assemblies, promoted by NEDD1, likely form a key platform for γ-TuRC binding. Our plant Augmin structures highlight the complexity of the multi-helical interactions stabilizing the Augmin hetero-octamer. Many of these interactions are highly conserved across plants, animals, and insects, contributing to the overall architecture of Augmin assembly shape. The multi-helical coiled-coil interactions of AUG1,2,3,4,5,6,7,8 make Augmins flexible, potentially allowing communication between their V-junction and extended regions in response to biochemical of mitotic phosphorylation cues. However, Augmin oligomerization appears to serve an as-yet unknown but critical function in forming the platform for γ-TuRC binding and MT branch nucleation (Fig. 8B). This behavior is reminiscent of other regulatory complexes, such as the crosslinker PRC1 or COPII coatomer proteins, which also undergo head-to-tail dimerization[36,37].

In conclusion, we have determined cryo-EM structures leading to a full de novo model for plant Augmin complex coiled-coil assembly that is verified with XLMS. We found that Augmin forms antiparallel dimers through conserved interfaces in its extended region, and that NEDD1 WD β-propeller domains directly bind the Augmin V-junction, close to its AUG6,7,8 MT binding site. Altogether, we present a model in which NEDD1 promotes Augmin's MT binding and oligomerization through downstream conformational effects that result in the recruitment of the γ-TuRC complex in MT branch formation.

## Methods

### Cloning, protein expression, and purification

*A. thaliana* (At) Augmin AUG1,2,3,4,5,6,7,8 subunits and other plant orthologs were aligned with metazoan and insect counterparts, revealing high conservation The C-terminal region (residues 338–644) of AUG8 contains a highly conserved α-helical domain, while its divergent, disordered N-terminal domain containing the MT binding region was excluded.

AUG2,6,7,8 sequences contained multiple plant-specific rare Arg codons absent in bacteria, necessitating codon optimization. Each AUG1,2,3,4,5,6,7,8 subunit ORF was cloned into pET3a vectors and assembled into polycistronic co-expression constructs (Fig. 1A and Supplementary Fig. 1F) with C-terminal 8Xhis tags on AUG5 (His) and AUG6-StrepII tag (Strep) (Supplementary Fig. 1A). AUG1,3,4,5 subunits were assembled into a polycistronic vector with AUG5 C-terminal GFP and His tags (Supplementary Fig. 1F).

The *At* NEDD1-WD β-propeller (residues 1–315) domain was cloned into pFastBac with strepII tag.

Augmin Polycistronic constructs were transformed into SoluBL21 (AMSBio) cells on ampicillin plates. Hetero-octameric (AUG,1,2,3,4,5,6,7,8) Augmin was overexpressed from overnight cultures diluted 1:200 into 2xYT media with 200 μg/mL ampicillin. Cells were induced with 0.5 mM IPTG at $OD_{600}$ 0.6, grown at 18 °C for 12–14 h, and harvested by centrifugation at 3000 rpm for 25 min. Pellets were resuspended in lysis buffer (50 mM HEPES pH 7.5, 200 mM KCl, 1 mM MgCl₂, 1 mM EGTA, 12 mM ß-mercaptoethanol, 10% (v/v) glycerol) with protease inhibitors (5 μg/mL leupeptin, pepstatin aprotinin, 0.1 mM PMSF, 5 μg/mL benzamidine, EDTA-free mini tablets (Sigma Aldrich). DNaseI was added and cells lysed by microfluidization. Clarified lysates (18,000 rpm, 20 min,

4 °C) were loaded onto recycled Ni-IDA columns (Macherey Nagel). Augmin was eluted with 200 mM Imidazole after washing. Assembly was verified by SDS-PAGE. Eluates were loaded onto Hitrap-QFF columns (Cytiva), and flow-through was further purified on Superose 6 16/60 columns in 50 mM HEPES pH 7.5, 200 mM KCl, 1 mM EGTA, 1 mM MgCL₂, 10% (v/v) glycerol. The hetero-tetrameric (AUG1,3,4,5) Augmin was purified similarly but eluted from Q-FF with ~400 mM KCl and gel filtered on Superdex 200 16/60 (Cytiva) in 50 mM HEPES pH 7.5, 300 mM KCl, 1 mM MgCl2, 1 mM EGTA, 5% (v/v) glycerol. Aliquots (3 mg/mL) supplemented with 15% (v/v) glycerol were snap-frozen.

*At* NEDD1-WD β-propeller expressed in Tni cells (Expression Systems, Davis CA, catalog number 94-002S)(1:20virus inoculum) 27 °C for 60–70 h. Cells (≥90% viability) were lysed in 50 mM HEPES pH 7.5, 250 mM KCl, 1 mM MgCl2, 1 mM EGTA, 5% (v/v) glycerol with protease inhibitors. NEDD1-WD was purified on regenerated Strep-Tactin columns (IBA) with 100 mM *D*-biotin elution and gel filtered on Superdex 200 16/60 (Supplementary Fig. 12) in lysis buffer. Aliquots (1 mg/mL) with 20% (v/v) glycerol were snap-frozen.

Purified full-length Augmin subunits were analyzed by LC-MS/MS (Taplin Facility). Gel bands were excised from 12% SDS-PAGE, digested, and identified peptides manually mapped.

### Mass photometry and multi-angle light scattering experiments

MP experiments were performed using a Refyen 1.0 mass photometer. Data were analyzed in Discover MP software using mass calibration provided by Refyen. Samples (Supplementary Figs. 1D, H and 12G, H) were crosslinked with a ~50–100-fold molar excess of glutaraldehyde, then diluted to 5–10 ng/mL in 12 μL 20 mM HEPES pH 7.0, 100 mM KCl. Non-crosslinked samples were analyzed without dilution in 50 mM HEPES pH 7.5, 150–300 mM KCl, 1 mM MgCl2, 1 mM EGTA.

SEC-MALS was performed using an HPLC system equipped with a Superdex 200 Increase 10/300 column in line with a Wyatt Mini-DAWN TREOS multi-angle light scattering detector and Optilab reflective index detector (Wyatt Technologies). AUG1,2,3,4,5,6,7,8 (200 μL, 1 mg/mL; Supplementary Fig. 1B) or AUG1,3,4,5 (200 μL, 1 mg/mL; Supplementary Fig. 1G) were injected and eluted in 50 mM HEPES pH 7.5, 200 mM KCl, 1 mM MgCl2, 1 mM EGTA. Molecular masses were determine using ASTRA software (Wyatt Technologies).

### Augmin and NEDD1-WD β-propeller binding experiments

Reconstitutions of hetero-octameric (AUG1,2,3,4,5,6,7,8) with NEDD1 was evaluated was by size exclusion chromatography on a Superose 6 5/150 column (Cytiva) in 50 mM HEPES pH 7.5, 150 mM KCl, 1 mM MGCl2, 1 mM EGTA, 5% (v/v) glycerol with 100 μL injections. AUG1,2,3,4,5,6,7,8 at 1 mg/mL was incubated with and without a tenfold molar excess of NEDD1-WD. Sucrose density gradient experiments were performed with AUG1,2,3,4,5,6,7,8 (~1 mg/mL) or AUG1,3,4,5 (~2 mg/mL) in the presence or absence of a fivefold molar excess of NEDD1-WD on 10–40% sucrose gradients in 50 mM HEPES pH 7.5, 150 mM KCl, 1 mM EGTA, 1 mM MgCl2 and fractions were evaluated using SDS PAGE (Supplementary Fig. 12C–H)

### Cryo-EM sample preparation and data collection

The hetero-octameric (AUG1,2,3,4,5,6,7,8) and Hetero-tetrameric (AUG1,3,4,5) Augmin samples were purified by size exclusion chromatography on Superdex 200 10/300 columns (Cytiva), concentrated to 1 mg/mL, and crosslinked with 200 nM BS3 (ThermoFisher) on ice for 2 h, followed by quenching with 10 μM Tris-HCl pH 8. AUG1,2,3,4,5,6,7,8-NEDD1-WD-β-propeller complexes were purified on a Superdex 200 16/60 column, concentrated to 3 mg/mL, crosslinked with 1 μM BS3 for 1 h, and quenched with 1 mM Tris-HCl pH 7.0. Buffers are described in the Supplementary Information.

Cryo-EM grids were prepared using a Leica EM GP2 with sensor blotting at 20 °C and 95% humidity, 5–10 s pre-blot, 5–8 s blot time, and 1.5–1.8 mm extra push. R 1.2/1.3 300 mesh grids (Quantifoil) were

used (Cu for AUG1,2,3,4,5,6,7,8 and AUG1,3,4,5; Au for AUG1,2,3,4,5,6,7,8-NEDD1-WD). Some AUG1,2,3,4,5,6,7,8 grids included 0.001% NP-40.

Grids were screened on a Glacios microscope (Thermo Fisher) with a K3 direct electron detector (Gatan) at 200 kV using SerialEM[38] at 11,000× and 45,000× magnifications using low dose conditions. High-resolution data were collected at 45,000× (0.44 Å/pixel) in Super-resolution mode) with SerialEM[38] and beam-image shift[39], at −0.6 to −1.8 μm defocus and 60 e−/Å² total dose over 75 frames.

## Cryo-EM single particle image processing

The full Augmin hetero-octamer (AUG1,2,3,4,5,6,7,8) structure (Supplementary Fig. 2, black arrows) was determined as follows: 5717 movies of hetero-octameric (AUG1,2,3,4,5,6,7,8) Augmin were motion-corrected (MotionCor2[40], 2× binned) and CTF-estimated (CTFfind[41]) in RELION[42]. Particles (20 million) were picked by LoG-based template-free picking (50–450 Å diameter) and extracted (200 pixels, 3.52 Å/pixel). After 2D classification in cryoSPARC[43], 714,446 particles were re-extracted in RELION[42] (300 pixels, 1.76 Å/pixel). Further 2D- and 3D-classification. Using ab initio model generation followed by refinement yielded a 9.6 Å map from 180,160 particles, 3DFlex[44] refinement in cryoSPARC[43] followed by 2D-classification and homogeneous refinement produced a 9.1 Å map based on Fourier shell correlation (FSC). Local resolution was calculated in PHENIX[45].

The Augmin V-junction-stem structure (closed state) structure (Supplementary Fig. 3) was determined as follows: Particles were re-centered on the V-junction and re-extracted (160 pixels, 1.76 Å/pixel). 2D-classification selected 440,976 particles, which were 3D-refined, CTF-refined, and polished[46] in RELION[42]. Duplicate-removed particles (223,355) were re-extracted (200 pixels, 1.76 Å/pixel), and ab initio model was generated and refined in cryoSPARC[43]. Heterogeneous refinement with three ab initio models yielded 107,694 particles in the closed state. Further 2D-classification and 3DFlex[44] refinement, followed by re-extraction, 2D- and 3D-refinement, produced a 7.3 Å closed V-junction map from 75,828 particles. A second dataset of hetero-octameric (AUG1,2,3,4,5,6,7,8) Augmin 11,256 movies was processed similarly, and particles from both datasets (200,853) were merged in cryoSPARC[43]. 2D, heterogeneous, non-uniform[47], and local refinement, combined with 3D variability analysis[48] and re-extraction (256 pixels, 1.76 Å/pixel), yielded a final 7.3 Å map from 18,243 particles (3DFlex[44] refinement, DeepEMhancer[49] sharpening).

The Augmin V-junction-stem (open state) structure (Supplementary Fig. 3, grey arrows) was determined as follows: 94,659 particles from heterogeneous refinement of the closed state were 3D refined, 2D classified, and re-extracted (200 pixels, 1.76 Å/pixel) to select 67,132 particles. Homogeneous, non-uniform[47], and 3DFlex[44] refinement, followed by 3D variability analysis[48] and individual frame reconstruction, selected 37,048 particles, 3DFlex[44] refinement, CTF refinement, and non-uniform refinement produced a 12 Å open V-junction map, after sharpened using DeepEMhancer[49].

The Antiparallel Augmin dimer structure (Supplementary Fig. 2, blue arrows) was determined as follows: 6578 particles from 2D classification were used to generate an initial dimer map in cryoSPARC[43], which was refined in RELION[42]. Particles were re-centered on the AUG1,3,4,5 overlap region and re-extracted (384 pixels, 0.88 Å/pixel). Homogeneous and non-uniform refinement with C2 symmetry, followed by 3DFlex[44] refinement, yielded an 8.1 Å map (B-factor sharpening).

The Augmin 1.5-mer structure (Supplementary Fig. 2, gray arrows) was determined as follows: 81,112 particles showing a second leg were separated by heterogeneous refinement, re-extracted (350 pixels, 1.76 Å/pixel), and 2D classified to select 11,903 particles. Ab initio reconstruction followed by homogeneous and non-uniform refinement produced a 12.4 Å 1.5-mer map.

The hetero-tetrameric (AUG1,3,4,5) Augmin extended region structure (Supplementary Fig. 4) *was determined as follows:* 10,830 movies were motion-corrected (MotionCor[40], 2× binned) and CTF-estimated (GCTF[50]). 15.5 million particles were picked by reference-free LoG picking (80–350 Å diameter) and extracted (100 pixels, 3.52 Å/pixel). 2D classification in cryoSPARC[43] selected 638,316 particles, which were re-extracted (200 pixels, 1.76 Å/pixel) and 3D refined in RELION[42], followed by CTF refinement and polishing of 387,892 particles. Iterative 3D refinement, 2D/3D classification, and re-extraction (500 pixel, 0.88 Å/pixel) yielded a 4.1 Å map from 247,267 particles. 3D classification selected 144,287 particles, which were re-extracted (512 pixels, 0.88 Å/pixel) and refined in RELION[42] and cryoSPARC[43] to produce a 3.9 Å map. Re-extraction (400 pixels, 0.88 Å/pixel), 2D classification, and iterative refinement in cryoSPARC[43] yielded a 3.5 Å map from 72,002 particles (3D variability analysis[48], 3DFlex[44] refinement, DeepEMhancer[49] sharpening). Focused local refinement of the tripod with a tight mask produced a 5.9 Å map.

The Augmin V-junction-NEDD1-WD-β-propeller structure (Supplementary Fig. 14) was determined as follows: 8813 movies were motion-corrected (motioncorr2, 2× binned), CTF-estimated (CTFFind3), and blob-picked (50–600 Å diameter) in cryoSPARC[43]. 2D classification of 10 million initial particles (200 pixels, 3.52 Å/pixel) selected 15,844 particles used for template picking, yielding 510,466 particles. 2D classification selected 39,064 particles, which were used to generate an ab initio model and 3D refined. Particles were re-extracted (250 pixels, 1.76 Å/pixel), and 2D classification selected 27,025 particles. Template-picking along with 3D refinements, classifications, re-extraction, and local refinement with 49000 particles (180 pixels, 1.76 Å/pixel) to solve 10 Å (DeepEMhancer[49] sharpening) map for Augmin (V-junction) with NEDD1-WD40.

## Model building and refinement

Model interpretation was limited to lower than reported resolutions due to resolution estimate inflation likely caused by map flexibility and low signal-to-noise of elongated particles, a trend observed in all published Augmin reconstructions[29–31]. Interpretation was based on visual map features rather than nominal resolutions. The extended region (AUG1,3,4,5) maps ranging from 3.5 to 4 Å were interpreted to 4 Å. The AUG1,3,4,5 tripod density and closed V-junction CH-domain maps at 5.9 Å were interpreted to 7.3 Å for the full Augmin model. Augmin (V-junction) with WD40 was built using closed CH-domain model with placement of WD40 domain (AlphaFold). Open CH-domain, 1.5-mer, and dimer maps were interpreted at 12, 15, and 10 Å, respectively.

Initial models were built by rigid body fitting AlphaFold3 predictions (Supplementary Fig. 8) for AUG1,3,4,5 and AUG1,2,3,4,5,6,7,8 into density using UCSF ChimeraX[51] and Coot[52]. The AUG3,5 fold-back zone and AUG2,6,7,8 regions were built into the 7.3 Å V-junction map by flexible fitting and manual adjustment based on AlphaFold3 secondary structure predictions. The open CH-domain conformation (Supplementary Fig. 6C) was modeled by rigid-body fitting into distinct CH-domain-like densities.

The 3.7 Å AUG1,3,4,5 map was used for de novo modeling of the interacting helical regions in the belly and legs, starting from the AlphaFold prediction. AUG3,5 N-terminal and C-terminal helical bundles and AUG1,4 C-termini were built into the 6 Å map using secondary structure element length, connectivity, and interactions as guides.

A full de novo AUG1,2,3,4,5,6,7,8 model was assembled by merging the above V-junction-stem and extended region models and fitting into the 8.1 Å consensus map using a 20 Å, 4-helix bundle in the central AUG3,5 (Supplementary Fig. 5A, B). The dimer model was built by rigid body fitting two copies of AUG1,2,3,4,5,6,7,8 into the dimer map. The AUG1,2,3,4,5,6,7,8-NEDD1-WD model (Supplementary Fig. 14B) was built by fitting the closed V-junction model and placing

**Table 1 | Cryo-EM data collection, processing, and model building**

| Data collection | Augmin datasets | | | | | | |
|---|---|---|---|---|---|---|---|
| Microscope | Thermofisher Glacios | | | | | | |
| Detector | Gatan K3 | | | | | | |
| Magnification | 45000× | | | | | | |
| Voltage (kV) | 200 | | | | | | |
| Electron exposure (e Å$^{-2}$) | 60 | | | | | | |
| Defocus range (μm) | −0.6 to −1.8 | | | | | | |
| Pixel size (Å) | 0.44 | | | | | | |
| **Reconstructions** | **Augmin V-junc (closed)** | **Augmin V-junc (open)** | **Augmin full** | **Augmin dimer** | **Augmin extended body** | **Augmin extended (Tripod)** | **Augmin V-junc/ NEDD1-WD** |
| **EMDB code** | EMD-49225 | EMD-49224 | – | EMD-49227 | EMD-49182 | EMD-49183 | EMD-72072 |
| Symmetry imposed | C1 | C1 | C1 | C2 | C1 | C1 | C1 |
| No. of final particle images | 18243 | 37048 | 123579 | 6578 | 72005 | 72005 | 49000 |
| Map resolution FSC threshold (Å) | 5.9 | 8.6 | 9 | 9.5 | 3.5 | 5.9 | 8 |
| Initial model used | AlphaFold multimer and AlphaFold2 prediction | | | | | | |
| Model resolution FSC threshold (Å) | 0.5 | | | | | | |
| **Model refinement PDB ID** | **9NBB** | **9NBA** | **–** | **9NBD** | **9NA8** | **9NA9** | **9PZM** |
| Model resolution (Å) | 8 | 12 | – | 12 | 5 | 7 | 15 |
| Model composition | | | | | | | |
| Chains | 6 | 6 | – | 8 | 4 | 4 | 7 |
| Non-hydrogen atoms | 11941 | 10976 | – | 22570 | 7364 | 4172 | 14095 |
| Protein | 1500 | 1375 | – | 2842 | 935 | 513 | 1785 |
| Ligand | 0 | 0 | – | 0 | 0 | 0 | 0 |
| **R.M.S. deviations** | | | | | | | |
| Bond length (Å) | 0.003 | 0.003 | – | 0.003 | 0.003 | 0.003 | 0.003 |
| Bond Angles (°) | 0.654 | 0.745 | – | 0.771 | 0.783 | 0.897 | 0.752 |
| **Model Validation** | | | | | | | |
| MolProbity score | 2.27 | 2.58 | – | 2.53 | 2.08 | 1.99 | 2.35 |
| Clash score | 24.58 | 36.07 | – | 34.69 | 14.51 | 24.08 | 25.49 |
| Poor rotamers (%) | 0 | 0 | – | 0 | 0 | 0.22 | 0 |
| **Ramachandran plot** | | | | | | | |
| Favored (%) | 94.35 | 90.17 | – | 91.53 | 95.67 | 97.41 | 92.83 |
| Allowed (%) | 5.65 | 9.76 | – | 8.40 | 4.33 | 2.59 | 7.11 |
| Disallowed (%) | 0 | 0.07 | – | 0.07 | 0 | 0 | 0.06 |

NEDD1-WD into the junction density. All models were subjected to real-space refinement in PHENIX[45] (Table 1).

**Crosslinking mass spectrometry (XL-MS) of Augmin assemblies**
AUG1,2,3,4,5,6,7,8 and AUG1,3,4,5 samples were crosslinked with 0.5–2 mM BS3 at 4 °C overnight, denatured (8 M urea), reduced (5 mM DTT), alkylated (15 mM IAA), quenched with DTT, digested with LysC (1:100 w:w ratio), diluted fourfold, digested with trypsin (1:50 w:w ratio), desalted (Sep-Pak C18), and vacuum-dried. Desalted peptides were fractionated via Superdex 30 10/300 GL gel filtration, then dried and stored at −80 °C. AUG1,3,4,5 fractions were analyzed using an UltiMate3000 UHPLC system coupled to a Q-Exactive HF-X Orbitrap. AUG1,2,3,4,5,6,7,8 fractions were analyzed with a Vanquish Neo UHPLC coupled to Orbitrap Ascend. Peptides were loaded onto PepMap 100 C18 column with Solvent A (0.1% formic acid in water) and Solvent B (0.1% formic acid in ACN). AUG1,3,4,5 peptides were separated using PepMap RSLC C18 column with a linear gradient from 5% to 25% solvent B over 100 min, then to 45% over 25 min, and then to 90% over 5 min. AUG1,2,3,4,5,6,7,8 peptides were separated with the same column with a linear gradient from 10.4% to 32.8% solvent B over 87 min, then to 44% over 5.5 min, and then to 76% over 2.5 min. For LC-MS/MS data acquisition, the Q-Exactive HF-X

performed MS1 scans at 120,000 resolution (350–1500 m/z), AGC target of $3 \times 10^6$, and 50 ms max IT. The top 10 precursors ($z = 3$–8) were isolated (1.4 m/z window) and fragmented using stepped NCE (30 ± 6). MS2 scans were at 60,000 resolution (200–2000 m/z), AGC target of $8 \times 10^3$, and 150 ms max IT. Dynamic exclusion was set to 45 s and in-source CID at 10 eV. For the Orbitrap Ascend, MS1 scans were at 240,000 resolution (380–2000 m/z), normalized AGC target of 150%, and 100 ms max IT. The top 20 precursors ($z = 3$–7 with 4–7 preferred) were isolated (1.4 m/z window) and fragmented (NCE 30 ± 6). MS2 scans were at 60,000 resolution (150–2000 m/z), normalized AGC target of 750%, and 250 ms max IT. Dynamic exclusion was set to 30 s and in-source CID at 10 eV.

RAW files were converted and recalibrated using the xiSEARCH preprocessing pipeline Python script. The recalibrated MGF files were then searched with xiSEARCH 1.8.6 for crosslinked peptides. Search parameters were used as following: MS1 mass tolerance, 6 ppm; MS2 mass tolerance, 10 ppm; allowed maximum number of missed cleavages, 4; minimum peptide length, 6. Crosslinks were searched based on the modifications at Lys and N-term (preferred), modifications at Ser, Thr, and Tyr were also allowed with lower priority. Carbamido-methylations (+ 57.021464 Da) on Cys were enforced as fixed modifications, oxidations (+ 15.99491463 Da) on Met were allowed as

variable modifications. FASTA files containing AUG1,3,4,5 or AUG1,2,3,4,5,6,7,8 were applied in the search separately. Search results were filtered in xiFDR 2.3.2 at the residue pair level to an FDR of 5%. The boost function was enabled between residue pairs, and the rest of the settings were default. The mzid file generated from xiFDR was uploaded onto xiVIEW and PDB files generated from cryo-EM study were imported for crosslink visualization. The same RAW files for AUG1,3,4,5 or AUG1,2,3,4,5,6,7,8 were also analyzed using FragPipe 22.0 to search for linear peptides. The proteins with the top 50 most peptide-spectral matches were used to generate alternative FASTA files for expanded crosslink searches that include contaminant proteins. The same xiSEARCH/xiFDR pipeline was applied to the expanded FASTA database search, and the respective mzid files were uploaded onto xiVIEW without importing the PDB files. Figure 5 and Supplementary Fig. 11D–F were generated by exporting the filtered crosslinks from xiVIEW and drawn in ChimeraX[51] with XMAS[53]. Connectograms in Supplementary Fig. 11G, H were generated with xiVIEW. Protein-protein interaction networks of AUG1,3,4,5 or AUG1,2,3,4,5,6,7,8 and contaminant proteins in Supplementary Fig. 12 were generated by xiVIEW.

### Mapping coevolutionary interactions through direct coupling analysis

Seed sequences and Hidden Markov model (HMM) profiles were obtained for the eight Augmin subunits from UniProt and Pfam (InterPro), respectively[54]. The HMM files containing the protein domains were used to search the UniProt database and generate MSA Fasta files, except for AUG1 for which there was no Pfam domain, and the complete seed sequence was used to create a custom made HMM to generate an MSA (Supplementary Table 3). Each MSA consisted of homologous protein sequences belonging to the same protein domain family from diverse species. Sequences with large gaps in MSAs (maximum continuous gap of 20% the length of sequence) were removed and the filtered MSAs were horizontally concatenated using the Progressive Paralog Matching algorithm (PPM)[55,56]. DCA was carried out on the paired MSAs to obtain the residue pairs across proteins with the largest coevolutionary coupling signals (Supplementary Table 1)[55]. To estimate the significance of classifying a DI pair as interacting, a hypergeometric test was performed and only the DI pairs with a $p$-value $\leq 0.2$ were retained (Supplementary Table 3), which were mapped onto the Augmin structure using Chimera (Supplementary Table 2). The $p$-values in Supplementary Table 3 were calculated using right tailed Hypergeometric test, testing for enrichment of true positive inter-domain DCA contacts among a given number of predicted DI pairs.

### NEDD1 coevolution and phylogenetic tree construction

Seed sequences for each of the eight Augmin subunits and NEDD1 were selected from *Homo sapiens*, *A. thaliana*, and *D. melanogaster* from UniProt. Each seed sequence was queried against the UniProt database using BLAST, and the top three hits per species, based on highest sequence identity and lowest $e$-value, were selected. This produced a total of nine representative sequences for each protein. These were aligned using Clustal Omega[57] and HMM profiles were constructed from the resulting alignments. The HMM profiles were used to search the UniProt database and generate MSAs for each of the nine proteins. The MSA of full NEDD1 sequences was filtered to remove sequences with maximum continuous gaps of 50% caused due to insertions and deletions. Using NCBI Taxa from ETE3[58], species were categorized from each protein MSA into taxonomic groups, which are represented in Fig. 7B. Subsequently, species that were present in all nine MSAs were identified, and a phylogenetic tree of these common species was constructed using the NCBI Taxonomy Browser and iTOL[59] presented in Supplementary Fig. 15A.

### Computational modeling of interaction between NEDD1 and Augmin

To investigate the potential interface between NEDD1 and the Augmin complex, seed sequences corresponding to NEDD1-WD β-propeller and the V-junction regions of AUG2,3,5,6,7,8 were taken from the V-junction of the Augmin hetero-octamer sequence regions. These sequences were used to generate MSAs for each subunit, which were then horizontally concatenated in six NEDD1-AUG combinations using PPM (Supplementary Table 4). DCA was performed on each paired MSA to identify coevolving residue pairs with high DI score (Supplementary Table 5). To model NEDD1 docking onto Augmin, the partial structure file from 9NBI containing only the V-junction region was modified in Chimera such that NEDD1-β-propeller was positioned approximately 60 Å away from the core of the Augmin assembly. From this, a topology and a.gro (Gromos87) file were prepared using the SMOG server[60].

The DI pairs from each NEDD1-AUG combination were combined and then ranked based on DI score and the DI pairs with a combined solvent accessible surface area of greater than 100 were retained[61]; the topology file was modified to include only the top 10 of these DI pairs under "pairs" and "exclusions" sections. Subsequently, coarse-grained molecular dynamics simulations were performed using GROMACS (version 4.5.4_sbm1.0) through a hybrid SBM + DCA (Structure-Based Modeling + Direct Coupling Analysis) framework. This approach used evolutionary constraints i.e., 5 out of the top 10 DI-ranked residue pairs (Supplementary Table 5) identified via DCA, to gradually guide NEDD1 toward the Augmin complex during the simulation, leading to a model presented in Fig. 7E.

### Reporting summary

Further information on research design is available in the Nature Portfolio Reporting Summary linked to this article.

## Data availability

Cryo-EM maps and models presented here will be available in the Electron Microscopy Database (EMDB) with the EMBD-IDs: EMD-49225, EMD-49224, EMD-49227, EMD-49182, EMD-49183, EMD-72072. The corresponding models are available at the Protein Data Bank (PDB) with the accession codes: 9NBB, 9NBA, 9NBD, 9NA8, 9NA9, and 9PZM, respectively. The mass spectrometry proteomics data have been deposited to the ProteomeXchange under accession code PXD065516. List of all crosslinks identified in restricted and expanded searches are provided as Supplementary Data 1. Source Data are provided as a Source Data file. Source data are provided with this paper.

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

## Acknowledgements

We would like to thank Dr. Ian Humphreys and Dr. David Baker (University of Washington, Seattle) for AlphaFold2 prediction of hetero-tetrameric and hetero-octameric Augmin assemblies. We thank the College of Biological Sciences and Departments of Plant Biology and Molecular Cellular Biology at UC-Davis for the generous support of this inter-departmental collaboration. We like to thank Dr Richard McKenney, Dr Gant Luxton (Molecular Cellular Biology UC-Davis) for the critical reading of this manuscript. Cryo-EM data were collected at the UC-Davis cryo-EM facility with support by the Molecular Cellular Biology department and College of Biological Sciences. We thank Dr Camille Scott at the UC-Davis high-performance computing facility for computational support for cryo-EM structure determination. J.A.B. acknowledges grant support from the National Institutes of Health (GM110283, GM158334). B.L. acknowledges grant support from the National Science Foundation (NSF/BSF-2416267 and NSF-2148207). S.D.F. acknowledges support from the National Institutes of Health (DP2-GM140926-01) and the Sloan Foundation.

## Author contributions

M.A. purified Augmin assemblies, carried out all biochemical studies, prepared XLMS samples, determined and refined all structures, built and refined all models, prepared figures, supervised the project, and co-wrote and co-revised manuscript. A.T. purified Augmin assemblies, prepared cryo-EM grids, collected cryo-EM data, and determined initial structures. Y.R.J.L. and Y.T. contributed equally to this work. Y.R.J.L. prepared bacterial polycistronic constructs for Augmin reconstitution and expression. Y.T. carried out XL-MS experiments, analyzed the data, and prepared figures. S.M. carried out DCA and coevolutionary analyses. F.G. assisted cryo-EM data collection and cryo-EM grid preparation. F.M. supervised and carried out coevolutionary analyses and revised manuscript. S.D.F. provided grant support, supervised and advised XL-MS studies, and revised the manuscript. B.L. provided grant support, supervised, and advised Augmin expression construct preparation and data interpretation. J.A.B. conceived the project, provided grant support, supervised and advised all co-authors, carried out biochemical experiments, prepared figures, wrote and revised manuscript.

## Competing interests

The authors declare no competing interests.
