## [Transparent Peer Review file · Nature Communications]

Cryo-EM structures of the Plant Augmin reveal coiled-coil assembly, antiparallel dimerization and NEDD1 binding

Corresponding Author: Dr Jawdat Al-Bassam

Version 0:

Reviewer comments:

Reviewer #1

(Remarks to the Author)

The nucleation of microtubule branching mediated by Augmin is essential for the construction of the mitotic spindle and cortical microtubules in plant cells. In this study, the authors elucidate the structural mechanism of the Augmin complex in *Arabidopsis thaliana* using Cryo-EM, cross-linking mass spectrometry, and biochemical analyses. The key findings are as follows:

1. The first high-resolution structural model of the plant Augmin complex, which differs from its counterparts in metazoans.
2. Demonstration of anti-parallel dimerization through conserved interaction sites
3. The dual CH domains located at the V-junction display either open or closed conformations, suggesting a mechanism for regulating binding to the microtubule lattice.
4. The NEDD1 WD40 domain binds directly to the V-junction of Augmin, supporting its dimerization and enhancing its interaction with microtubules.

The authors also suggest that NEDD1 and the CH domains of Augmin create a platform for the recruitment of γ -TuRC through cooperative binding.

This study represents a significant advancement in understanding the role of the Augmin complex in microtubule nucleation. The insights derived regarding its structure are valuable and contribute significantly to the field. It is recommended that this manuscript be considered for publication in Nature Communications, provided that the following concerns are addressed:

Major points:

1. Validate the structural models using mutagenesis analyses, especially for the Augmin dimer and the Augmin/NEDD1-WD40 complex. Additionally, it is preferable to evaluate these mutants *in vivo* for comprehensive results.

Minor points:

1. Line 79: The referenced study appears to indicate that augmin recruits NEDD1 to microtubule, rather than the reverse.
2. A recent study by Gao et al. (Nat Commun, 2025) suggests that the region corresponding to the tripod of plant Augmin interacts with the NEDD1/GCP3/MZT1 module. It raises the question of whether the tripod region of plant Augmin can be integrated into the proposed NEDD1/GCP3/MZT1/Augmin model as predicted by AlphaFold.
3. The direct interaction between the NEDD1 WD40 domain and the V-junction of Augmin is noteworthy, particularly considering that human Augmin does not exhibit binding to the NEDD1 WD40 domain in pulldown assays, as reported by Zhang et al. (J Cell Biol, 2022). Could the authors provide further insight into a potential explanation for this discrepancy?

Reviewer #2

(Remarks to the Author)

By examining *in vitro* reconstituted *Arabidopsis* Augmin complex using cryoEM and crosslinking mass spectrometry, Ashaduzzaman et al. proposed a model elucidating how the cooperativity between Augmin complex and NEDD1 may generate a platform to recruit gamma-tubulin ring complex and activate branched microtubule nucleation. Overall, the experiments are well-designed, and the data interpretation is appropriate. This work is suitable for publication in Nature

Communications after addressing some concerns.

Major concerns

1. It is not convincing that the high electron density spot at the V-junction is potentially due to the association with nucleic acid. Can the authors provide some supporting evidence? The proposed beta-barrel from bacteria should not contribute to the strong electron density. In particular, it is puzzling why this strong signal in 2D class averages does not lead to a well-defined density in the 3D reconstruction.
2. Line 205-207, if AUG2,6,7,8 move upward and AUG3,5 in the base also move upward, where will be the pivot point for the movement?
3. In the crosslinking mass spectrometry, the authors suggest the hetero-dimeric assembly is more dynamic than the hetero-octamer. However, I do not think this statement is well-supported by their data. It is particularly unclear why the authors use the 30 Å cutoff.
4. How does the co-expression of the NEDD1-WD40 beta-propeller change the distribution of 2D class averages?
5. If a bacterial beta-barrel domain is copurified with the Augmin complex, the authors should be able to identify the unknown protein binder in their crosslinking mass spectrometry. Can the author perform the proposed mass spectrometry analysis? If not, what are the possible reasons?

Minor concerns

1. The section 'cryo-EM imaging of Augmin reveal dynamic architecture and higher order oligomers' is confusing. It might be helpful to add a cartoon to show the referred structural domains.
2. Figure 1Ev is not mentioned in the text.
3. It will be informative to show the ratio of each 2D-class average shown in Figure 1E.
4. Line 263. I think 'opposite topologies' is a confusing term.
5. The section "Augmin hetero-octameric structure stabilized by long and short-range coiled-coil interactions" is repetitive. I suggest the authors revise these paragraphs.
6. Figure 5 is very confusing with all the colors.

Reviewer #3

(Remarks to the Author)

In the paper entitled "Cryo-EM structures of the Plant Augmin reveal its intertwined coiled-coil assembly, antiparallel dimerization, and NEDD1 binding mechanisms," the authors have determined cryo-EM structures that lead to a complete de novo model of the coiled-coil assembly of the plant Augmin complex. They employed XL-MS approaches to validate these findings. In their models, Augmin forms antiparallel dimers through conserved interfaces in its extended region. The authors also describe how the WD40 β -propeller domains of NEDD1 directly bind to the Augmin V-junction, near the AUG6/7/8 microtubule (MT) binding site.

Altogether, this paper presents a model in which NEDD1 promotes Augmin's MT binding and oligomerization through downstream conformational effects, ultimately resulting in the recruitment of the γ -TuRC complex during MT branch formation.

The data presented are of high quality and of particular interest, especially given the cryo-EM analysis of such elongated particles. The findings are well supported by clear figures and an appreciated supplementary video. Although the resolution achieved by cryo-EM was moderate in some regions (10–20 Å), the integration of multiple techniques (such as XL-MS and computational modeling) enabled the authors to propose a robust model where NEDD1 facilitates Augmin's MT interactions. As I am not an expert in cryo-EM, my primary focus was on the biochemical and cross-linking experiments. However, the cross-linking data are difficult to evaluate since the raw files are not available. Additionally, the exported search results do not align with the manuscript, and thus a thorough revision of this section is necessary prior to publication.

Points to be addressed by the authors:

- The authors state that they identified 209 cross-links (XLS) within the Augmin hetero-octamer. However, after extracting all identified XLS from the provided .csv file, and removing duplicates and decoys, I arrive at a total of 191 unique XLS. Could the authors clarify the origin of the reported number 209?
- The two exported datasets differ significantly: one appears to be a raw export, including decoy and reverse database results, while the other seems to be a pre-processed version. Could the authors please provide a detailed description of the data processing pipeline and ensure consistent exports for both the tetrameric and octameric forms?
- In Figure S11, the model contains significantly fewer XLS than the number of XLS reportedly identified, despite the caption stating: "Augmin model presented in ribbon format with the observed crosslinks generated in tube format." The model clearly does not contain ~200 or even ~100 crosslinks. It appears that long-distance XLS (>30 Å) were excluded from the visualization, as none are visible. If this is the case, this filtering criterion must be clearly stated in the figure caption.
- The authors analyzed how many crosslinks from the hetero-octameric Augmin complex (AUG1–8) can be explained when allowing crosslinks to span across assemblies in the anti-parallel dimer model. While this is a valuable analysis, why is it limited to the octameric assembly, when the tetrameric form (Figure S11E) also shows four homo-inter XLS, compared to only two in the octameric form? This comparison should be expanded or explained.
- Line 367: The statement that "many XLS are found, suggesting the region is folded but flexible," is not necessarily accurate. A high number of XLS may also result from an unfolded region that is spatially close due to the folding and stabilization of other parts of the complex. This interpretation should be reconsidered or rephrased.
- Line 382: The authors claim that "plots of our detailed XL-MS crosslinks on the de novo Augmin model are fully consistent with the critical and distinguishing features of our structural model." However, this seems overstated, given that many XLS exceed the expected BS3 crosslinking distance (typically around 35 Å, with some reported exceeding 100 Å). This

statement should be tempered and clarified, possibly acknowledging model limitations or alternative explanations.

- Since “mass spectrometry also identified several contaminating β -barrel-containing proteins” that copurify with Augmin through multiple steps (as mentioned in the Supplementary list), these proteins should have been included in the database search used for the XL-MS analysis. Identifying binding sites between these β -barrel proteins and Augmin could have been helpful for structural interpretation. If insufficient XLs were obtained from co-purified complexes, the authors might consider performing a dedicated XL-MS experiment on the complex formed between the full Augmin hetero-octamer (AUG1–8) and the NEDD1-WD40 β -propeller domain.

Minor issues to be addressed:

Line 129: Figure 1A does not illustrate the presence or absence of conserved α -helices; it only shows the construct length and the position of the tags.

Similarly, Line 139: Figure 1A does not provide any relevant information regarding the statement “Mass spectrometry confirms each Augmin subunit is purified in the assembly and that the AUG6 C-terminal region is prone to degradation with multiple polypeptide bands.” This should be clarified or corrected.

Line 141: The manuscript references mass spectrometry data with “(Supplementary list file),” but the actual data are missing. Please ensure that this file is included.

Line 348: It is unclear which complex (tetramer or octamer) the reported values “50% (105 out of 209)” and “70% (80 out of 115)” refer to. This should be explicitly stated.

Version 1:

Reviewer comments:

Reviewer #1

(Remarks to the Author)

Regarding my minor point #1, I continue to believe that the authors should revise the text in line 82. Additionally, identifying RplB as the globular density in the cryo-EM map is not very convincing. If RplB tightly binds to augmin, it should be possible to detect the RplB band on SDS-PAGE and easily identify it by traditional mass spectrometry along with augmin subunits (line 140-141).

Reviewer #2

(Remarks to the Author)

The authors have addressed my questions. I have no further comments.

Reviewer #3

(Remarks to the Author)

I would like to thank the authors for their efforts in addressing my previous remarks and for improving their data analysis, particularly concerning the XL-MS part of their paper entitled “Cryo-EM structures of the Plant Augmin reveal its intertwined coiled-coil assembly, an Gparallel dimerization, and NEDD1 binding mechanisms.” I also appreciate their willingness to make the raw data easily accessible, thereby ensuring transparency and open access to their research.

After this revision, the XL-MS section is now much clearer, easier to follow, and aligns better with the structural models proposed for the Augmin assemblies.

Nevertheless, before this work can be published—as it indeed deserves—there remain some discrepancies between the raw data and the data presented in the paper that must be resolved. Specifically, when examining the raw exports from the XiSearch search engine after FDR validation (i.e., links_xiFDR2.3.2.csv files), I could not find several cross-links (XLs) reported in the paper’s dataset (Dataset S1 Source file XLS). Neither the total number of XLs nor the specific XL amino acid pairs match.

Initially, I considered that redundancy might explain the differences; however, after confirming that there were no duplicates or decoys, discrepancies persisted. For example, in the tetrameric XL dataset, the paper reports 216 cross-links (consistent with the text), whereas the T8_15_combined_Links_xiFDR2.3.2.csv file in the PRIDE repository contains 277. Similarly, in the octameric XL experiment, 115 cross-links are reported in the paper but 172 appear in the raw export from PRIDE. If the authors applied additional filters beyond the FDR, these criteria should be explicitly stated and justified.

Moreover, despite some cross-links being absent from the raw export, others appear in the dataset presented in the paper but are not found in the original data. When comparing the octameric XL lists, I identified 26 cross-links that are present in the paper but absent from the raw export. While it is understandable that certain XLs might be excluded due to stricter filtering (which should also be explained), it is unclear how non-identified XLs could appear in the published dataset. The authors should correct this inconsistency.

Minor Comments

Lines 391–394:

“Because the dimer model lacks the V-junction, only 59 of the 115 cross-links could be mapped to the dimer model, but of those 59, 32 had even shorter $C\alpha$ – $C\alpha$ distances in the anti-parallel dimer model, suggesting the presence of Augmin assemblies in the anti-parallel dimer state in solution (Figure 5C; Figure S11C,F).”

This sentence is overly result-oriented. The fact that a cross-link exhibits a shorter $C\alpha$ – $C\alpha$ distance in one model compared

to another does not necessarily imply stronger support for that model, especially when both fall within the range of a cross-linker. The probability of forming a cross-link (crosslinkability) is determined by multiple structural and dynamic factors. For a discussion of such criteria, please refer to:

Filella-Merce, I. et al. (2020). "Quantitative Structural Interpretation of Protein Crosslinks."

Version 2:

Reviewer comments:

Reviewer #1

(Remarks to the Author)

The authors have addressed my questions. I have no further comments.

Reviewer #3

(Remarks to the Author)

I would like to thank the authors for the constructive exchanges regarding their manuscript.

The authors have satisfactorily addressed my questions, and I have no further comments.

To Editor and Reviewers,

We would like to submit a revised version of our manuscript "Cryo-EM structures of the Plant Augmin reveal its intertwined coiled-coil assembly, antiparallel dimerization and NEDD1 binding mechanisms" to *Nature Communications*.

We would like to thank the reviewers for their helpful suggestions. In the revised manuscript we include the following major revisions:

- 1) We present a revised Crosslinking mass spectrometry (XLMS) analysis of bacterial protein co-purifying with Augmin assemblies and identify the RplB protein as the potential β -barrel protein binding to the Augmin V-junction. Using this information we have revised the presentation and discussion of the density bound to the Augmin V-junction. We have also utilized a variety of methods to try to produce more informed theories about its potential contents.
- 2) We have revised the presentation of the Crosslinking mass spectrometry (XL-MS) data to address concerns of the reviewers and clarified much of the analyses and fully rewrote all sections associated with XL-MS.
- 3) We present a new coevolutionary analysis of Augmin subunits and NEDD1 across plants, chordates, insects revealing the high degree of conservation of each of the eight Augmin subunits and NEDD1. We also present a direct coupling analysis (DCA) across all eight Augmin subunits to identify physical (direct information) pairings of changed amino acids in relationship to the structural model presented. The DCA analysis provides detailed multi-subunit distance interfaces that independently validate with the structural model of the Augmin assembly presented. The DCA analysis provides independent validation for the Augmin binding site for NEDD1 WD β -propellor being the V-junction region.
- 4) We present an improved cryo-EM single particle structure of the AUG1,2,3,4,5,6,7,8 - NEDD1 WD complex. This improved cryo-EM map shows improved density features of the NEDD1-WD at this junction. We also carried out DCA analysis of the NEDD1 WD β -propellor to determine its interface with the Augmin V-junction composed of AUG3,5 and AUG2,6,7,8. The DCA analysis provides further validation of the NEDD1-WD β -propellor binding site onto the Augmin V-junction in support of the biochemical and structural analysis

Below is a point by point response to the reviewer comments

REVIEWER COMMENTS

Reviewer #1 (Remarks to the Author):

The nucleation of microtubule branching mediated by Augmin is essential for the construction of the mitotic spindle and cortical microtubules in plant cells. In this study, the authors elucidate the structural mechanism of the Augmin complex in *Arabidopsis thaliana* using Cryo-EM, cross-linking mass spectrometry, and biochemical analyses. The key findings are as follows:

1. The first high-resolution structural model of the plant Augmin complex, which differs from its counterparts in metazoans.
2. Demonstration of anti-parallel dimerization through conserved interaction sites
3. The dual CH domains located at the V-junction display either open or closed conformations, suggesting a mechanism for regulating binding to the microtubule lattice.
4. The NEDD1 WD40 domain binds directly to the V-junction of Augmin, supporting its dimerization and enhancing its interaction with microtubules.

The authors also suggest that NEDD1 and the CH domains of Augmin create a platform for the recruitment of γ -TuRC through cooperative binding.

This study represents a significant advancement in understanding the role of the Augmin complex in microtubule nucleation. The insights derived regarding its structure are valuable and contribute significantly to the field. It is recommended that this manuscript be considered for publication in Nature Communications, provided that the following concerns are addressed:

We thank the reviewer for their positive assessment about our work. We provide a detailed point-by-point response to the reviewer's revision suggestions.

Major points:

1. Validate the structural models using mutagenesis analyses, especially for the Augmin dimer and the Augmin/NEDD1-WD40 complex. Additionally, it is preferable to evaluate these mutants *in vivo* for comprehensive results.

We thank the reviewer for this suggestion. Although this suggestion is fair, because of the substantial amount of work and cost associated with preparing, and isolating mutant augmin assemblies, the inclusion of these experiments would substantially delay the publication of the current results. We think these studies would be more suitable for a separate study that focuses on the structure in its biological context.

Minor points:

1. Line 79: The referenced study appears to indicate that augmin recruits NEDD1 to microtubule, rather than the reverse.

There are multiple experiments presented in Zhang et al 2022 that show that NEDD1 and Augmin have their own independent interaction with microtubules. We have revised the text

to explicitly indicate that both Augmin and NEDD1 have their own interactions with microtubules and their combined interaction with each other in the complex presented in this manuscript shows how they may form a more stable interface with microtubule lattice suitable for anchoring the microtubule branch nucleation complex mediated by the gamma-tubulin ring complex.

2. A recent study by Gao et al. (Nat Commun, 2025) suggests that the region corresponding to the tripod of plant Augmin interacts with the NEDD1/GCP3/MZT1 module. It raises the question of whether the tripod region of plant Augmin can be integrated into the proposed NEDD1/GCP3/MZT1/Augmin model as predicted by AlphaFold.

Two recent studies have addressed the NEDD1- helical domain binding to the g-TURC (Gao et al 2025 and Hugo Muñoz-Hernández et al 2025). Both studies show that the NEDD1 C-terminus forms a four-helix bundle that binds the g-TURC. We have revised the text to add in the potential binding of the AUG1,3,4,5 tripod to NEDD1/MTZ1/GCP3. However, the AlphaFold observation presented does not predict the accurate fold we describe here for the AUG1,3,4,5 foot in binding this assembly and thus the interface may be inaccurate as presented in those papers.

3. The direct interaction between the NEDD1 WD40 domain and the V-junction of Augmin is noteworthy, particularly considering that human Augmin does not exhibit binding to the NEDD1 WD40 domain in pulldown assays, as reported by Zhang et al. (J Cell Biol, 2022). Could the authors provide further insight into a potential explanation for this discrepancy?

The NEDD1 WD was shown to bind microtubules in the Zhang et al 2022 study. We believe it is possible that in the human system the affinity of the NEDD1-WD for Augmin is enhanced upon binding microtubules. It is possible microtubules play a crucial role in assembling the Augmin and NEDD1 complex. WE have revised the text to add this possibility.

Reviewer #2 (Remarks to the Author):

By examining in vitro reconstituted Arabidopsis Augmin complex using cryoEM and crosslinking mass spectrometry, Ashaduzzaman et al. proposed a model elucidating how the cooperativity between Augmin complex and NEDD1 may generate a platform to recruit gamma-tubulin ring complex and activate branched microtubule nucleation. Overall, the experiments are well-designed, and the data interpretation is appropriate. This work is suitable for publication in Nature Communications after addressing some concerns.

Major concerns

1. It is not convincing that the high electron density spot at the V-junction is potentially due to the association with nucleic acid. Can the authors provide some supporting evidence? The proposed beta-barrel from bacteria should not contribute to the strong electron density. In particular, it is puzzling why this strong signal in 2D class averages does not lead to a well-defined density in the 3D reconstruction.

The nature of the high electron density at V-junction has puzzled us too. We initially labeled its association with nucleic acid due to its 3-4-fold higher intensity than the protein material of the V-junction onto which it binds. Based on a reanalysis of our mass spec data collected of the isolated Augmin complexes with an expanded list of proteins in the search database, we now believe the density belongs to RplB, a ribosomal protein contaminant from *E. coli* that co-precipitates with Augmin (see response to #5). It is possible that because *E. coli* proteins tend to be quite rigid and stable (at least in comparison to Augmin which has an elongated and complex structure), that its electron density is less diffuse leading to the observed difference in intensity.

The identified protein is a ribosomal β -barrel protein with affinity to rRNA. If it co-elutes with rRNA, this could also explain why the density has 3-4-fold higher intensity than the augmin protein subunits it is bound to. It is noteworthy, however, that this density lies at the binding site for NEDD1-WD β -propeller, which is absent in the AUG1,2,3,4,5,6,7,8 expression system. However, reconstitution of NEDD1 led to a more uniform electron density at the V-junction in the AUG1,2,3,4,5,6,7,8-NEDD1 WD 2D-class averages and their Cryo-EM map (Figure 6 and Figure S13), suggesting that NEDD1 likely can replace RplB in vitro.

2. Line 205-207, if AUG2,6,7,8 move upward and AUG3,5 in the base also move upward, where will be the pivot point for the movement?

We believe Augmin has continuous coordinated movement and thus it is sparsely distributed across throughout the structure of the V-junction as evident from our flexibility analysis. See Figure S7 which provides details into the flexibility of the V-junction.

3. In the crosslinking mass spectrometry, the authors suggest the hetero-dimeric assembly is more dynamic than the hetero-octamer. However, I do not think this statement is well-supported by their data. It is particularly unclear why the authors use the 30 Å cutoff.

The XL-MS section has been rewritten in the revised manuscript. The original reason for stating that the anti-parallel dimer was more dynamic was mostly motivated by the observation that we could create a full reconstruction for the hetero-octamer (containing the V-junction) but the anti-parallel dimer model only contains the extended region (and without density we can assign to AUG2,6,7,8). It is of course possible that this is due to various technical reasons and not due to the anti-parallel dimer being more dynamic, so we have removed statements suggesting as much.

The 30 Å cut-off is based on the length of the spacer-arm in BS3. The fully extended BS3 has a maximum arm of 11.4 Å. However, because lysine itself has a long side-chain and reactions to the crosslinker occur at the distal amine, the theoretical distance between alpha carbons on crosslinked lysines is 24 Å if the chains are distended. However, due to the flexibility of proteins, various authors have considered Ca-Ca distances between 30 – 35 Å to be the range that would make an identified crosslink “consistent” with a given structural model.

As discussed in the text, the hetero-tetrameric assembly (AUG1,3,4,5) has 56% crosslinks with C α -C α distance within 35 Å, whereas the hetero-octameric assembly has 74% crosslinks fell within this range. On the other hand, in the hetero-tetrameric assembly, we found 42 “long” crosslinks (> 100 Å, 19%) connecting distal regions (tripod, belly and stem), whereas there were only 16 long crosslinks found in the hetero-octameric assembly. Hence, this constitutes evidence that the hetero-tetramer is more dynamic, which is also consistent with the cryo-EM itself. So this is an argument that we now emphasize more in the revised manuscript.

4. How does the co-expression of the NEDD1-WD40 beta-propeller change the distribution of 2D class averages?

The NEDD1-WD β -propeller protein was not soluble in the same expression system as AUG1,2,3,4,5,6,7,8. Thus, we had to reconstitute AUG1,2,3,4,5,6,7,8 (full Augmin) and NEDD1-WD β -propeller *in vitro*. In 2D class averages, we see a larger mass with more uniform signal for AUG1,2,3,4,5,6,7,8-NEDD1 WD β -propeller complex, compared to higher intensity smaller spot in the AUG1,2,3,4,5,6,7,8 alone. Also, we observed a larger proportion of Augmin anti-parallel dimer formation than Augmin alone which is experimentally supported by our mass photometry results. We have revised the text to include these details.

5. If a bacterial beta-barrel domain is copurified with the Augmin complex, the authors should be able to identify the unknown protein binder in their crosslinking mass spectrometry. Can the author perform the proposed mass spectrometry analysis? If not, what are the possible reasons?

Thank the reviewer for the suggestion. To begin, we reanalyzed the fractions (that we gathered when we were fractionating Augmin peptides) for linear peptides against the whole *E. coli* proteome, and identified many *E. coli* protein contaminants comingled with Augmin hetero-octamer. Proteins with very high spectral counts include chaperones GroEL and DnaK, DeaD, and Pnp, but also many RNAP and Ribosome subunits. Although suggestive that these proteins interact with Augmin, co-precipitation can occur for many reasons, including non-specific binding to resin. Using these results however, we re-performed a crosslinking search on the same mass spectra with a limited search database containing the top 50 proteins (42 contaminants for hetero-octamer and 46 contaminants for hetero-tetramer). The results show indeed that several *E. coli* proteins bind to Augmin, including Pnp, DnaK, and several ribosomal proteins (RpmA, RplB, RplD, RpsD). The XiView visualization of this map is shown in the paper as Figure S12. Notably, RplB's crosslinks target it to the V-junction, which is evidenced by two crosslinks, to K401 and K747 on AUG5 (which are both located at the V-junction of the Augmin complex). Hence, we tentatively assign this density to RplB (see above comment regarding unknown V-junction density). We do think that the evidence we can present so far provide sufficient justification that Augmin does copurify with *E. coli* proteins as contaminants, and that this can explain the presence of an additional density found in our cryo-EM images.

Minor concerns

1. The section ‘cryo-EM imaging of Augmin reveal dynamic architecture and higher order oligomers’ is confusing. It might be helpful to add a cartoon to show the referred structural domains.

This is a helpful suggestion, and we added a cartoon diagram in the revised figure.

2. Figure 1Ev is not mentioned in the text.

We thank the reviewer for catching the error. The reference is for Figure 1E. We fixed this in the revised manuscript.

3. It will be informative to show the ratio of each 2D-class average shown in Figure 1E.

We added the percentile information for each of the class averages in Figure 1E. –

4. Line 263. I think ‘opposite topologies’ is a confusing term.

We have revised the sentence to improve clarity.

5. The section “Augmin hetero-octameric structure stabilized by long and short-range coiled-coil interactions” is repetitive. I suggest the authors revise these paragraphs.

We revised the paragraph to focus down the details and remove the repetition.

6. Figure 5 is very confusing with all the colors.

We have revised Figure 5 to dim the colors of the Augmin subunits. We a dimmer tone colored augmin complex in the revised figure 5, which is necessary for the presentation otherwise it is hard to see which subunit interacts in XLMS data.

Reviewer #3 (Remarks to the Author):

In the paper entitled “Cryo-EM structures of the Plant Augmin reveal its intertwined coiled-coil assembly, antiparallel dimerization, and NEDD1 binding mechanisms,” the authors have determined cryo-EM structures that lead to a complete de novo model of the coiled-coil assembly of the plant Augmin complex. They employed XL-MS approaches to validate these findings. In their models, Augmin forms antiparallel dimers through conserved interfaces in its extended region. The authors also describe how the WD40 β -propeller domains of NEDD1 directly bind to the Augmin V-junction, near the AUG6/7/8 microtubule (MT) binding site. Altogether, this paper presents a model in which NEDD1 promotes Augmin’s MT binding and oligomerization through downstream conformational effects, ultimately resulting in the recruitment of the γ -TuRC complex during MT branch formation.

The data presented are of high quality and of particular interest, especially given the cryo-EM analysis of such elongated particles. The findings are well supported by clear figures and an appreciated supplementary video. Although the resolution achieved by cryo-EM was moderate in some regions (10–20 Å), the integration of multiple techniques (such as XL-MS and computational modeling) enabled the authors to propose a robust model where NEDD1 facilitates Augmin’s MT interactions.

As I am not an expert in cryo-EM, my primary focus was on the biochemical and cross-linking

experiments. However, the cross-linking data are difficult to evaluate since the raw files are not available. Additionally, the exported search results do not align with the manuscript, and thus a thorough revision of this section is necessary prior to publication.

We thank the reviewer for their helpful comments.

We had originally submitted our raw mass spec data to PRIDE, under the accession PXD060855. We apologize for the inconvenience that the reviewer was not able to locate them; this must have been due to a miscommunication whereby we had provided the PRIDE reviewer login details upon submission, but the relevant information may have not been accessible to the reviewer.

In any event, we have actually created a new PRIDE submission accession under the accession PXD065516. The access details are directly in the manuscript, but also reproduced here for your convenience:

Token: uleponAOr1sH

Username: reviewer_pxd065516@ebi.ac.uk

Password: cxd2zPo52d6l

The reason being is that in the time since our manuscript was originally submitted, a new set of data reporting standards have been described for XL-MS data (see [10.1016/j.mcpro.2025.101024](https://doi.org/10.1016/j.mcpro.2025.101024)) which recommend providing mzIdentML files to facilitate data reuse and reanalysis. Our new PRIDE submission adheres to this standard.

The reviewer is correct also that search results (originally) did not align with the manuscript. We discovered that there was an error from using Chimera to visualize crosslinks and XiView because the PDBs have missing residues (where there is missing density) but all crosslinking data assign residue numbering based on the full-length sequences. This has since been rectified in the revised Figure 5. Now, the crosslinks shown in the Figure 5 and Figure S11-S12, in the supplemental excel file Data S1 (which provides all unique residue pairs), and in the search results (which is on PRIDE) are all consistent.

Points to be addressed by the authors:

• **The authors state that they identified 209 cross-links (XLs) within the Augmin hetero-octamer. However, after extracting all identified XLs from the provided .csv file, and removing duplicates and decoys, I arrive at a total of 191 unique XLs. Could the authors clarify the origin of the reported number 209?**

We have since reanalyzed all the crosslinking data, and fixed a serious error in which there was a mismatch between the residue numbering between the PDB and XiFDR/XiView. The new, final, counts are 216 unique residue-pairs in the hetero-tetramer and 115 unique residue-pairs in the hetero-octamer. These 216 and 115 crosslinks are enumerated in Data S1. We have ensured that decoys and duplicates are removed.

- **The two exported datasets differ significantly: one appears to be a raw export, including decoy and reverse database results, while the other seems to be a pre-processed version. Could the authors please provide a detailed description of the data processing pipeline and ensure consistent exports for both the tetrameric and octameric forms?**

We thank the reviewer for pointing out the confusion. Our supplemental data file now provides filtered unique residue-pairs, which is the form that would probably be the easiest for other researchers to understand and work with.

Here is the detailed description of the data processing pipeline, reproduced from the text: RAW files were first converted and recalibrated using the xiSEARCH preprocessing pipeline python script. The recalibrated MGF files were then searched with xiSEARCH 1.8.6 for crosslinked peptides. Search parameters were used as following: MS1 mass tolerance, 6 ppm; MS2 mass tolerance, 10 ppm; allowed maximum number of missed cleavages, 4; minimum peptide length, 6. Crosslinks were searched based on the modifications at Lys and N-term (preferred), modifications at Ser, Thr, and Tyr were also allowed with lower priority. Carbamidomethylations (+57.021464 Da) on Cys were enforced as fixed modifications, oxidations (+15.99491463 Da) on Met were allowed as variable modifications. Search results were filtered in xiFDR 2.3.2 at the residue pair level to an FDR of 5%. The boost function was enabled between residue pairs, and the rest of the settings were default.

- **In Figure S11, the model contains significantly fewer XLS than the number of XLS reportedly identified, despite the caption stating: “Augmin model presented in ribbon format with the observed crosslinks generated in tube format.” The model clearly does not contain ~200 or even ~100 crosslinks. It appears that long-distance XLS (>30 Å) were excluded from the visualization, as none are visible. If this is the case, this filtering criterion must be clearly stated in the figure caption.**

The qualitative inconsistency that the reviewer is commenting on was another unfortunate consequence of the misnumbering error described before and now fixed. In the current version of the manuscript, we are much more explicit that we are only showing the crosslinks that fall within 35 Å. To ensure that this is clear, it is mentioned in a piece of text embedded in the figure itself (e.g., “216 crosslinks, 122 shown (<35 Å)”), in the figure legend, (“In total, 216 unique residue pairs were identified, of which the 122 with C α -C α distances less than 35 Å are displayed”), and in how it is discussed in the text.

- **The authors analyzed how many crosslinks from the hetero-octameric Augmin complex (AUG1–8) can be explained when allowing crosslinks to span across assemblies in the anti-parallel dimer model. While this is a valuable analysis, why is it limited to the octameric assembly, when the tetrameric form (Figure S11E) also shows four homo-inter XLS, compared to only two in the octameric form? This comparison should be expanded or explained.**

The single particle cryo-EM imaging suggests that the AUG1,3,4,5 assembly does not form anti-parallel dimers, while the AUG1,2,3,4,5,6,7,8 assembly forms significant dimers. The structural explanation for this stems from the AUG3,5-foldback zone being disordered as in AUG1,3,4,5 but not AUG1,2,3,4,5,6,7,8. Thus, we wanted to use the XLMS data to explain these the organization of this antiparallel dimer.

• **Line 367:** The statement that “many XLs are found, suggesting the region is folded but flexible,” is not necessarily accurate. A high number of XLs may also result from an unfolded region that is spatially close due to the folding and stabilization of other parts of the complex. This interpretation should be reconsidered or rephrased.

We thank for the reviewer pointing out the statement. This statement has been deleted, and indeed the whole XL-MS section has been rewritten in the revised manuscript.

• **Line 382:** The authors claim that “plots of our detailed XL-MS crosslinks on the de novo Augmin model are fully consistent with the critical and distinguishing features of our structural model.” However, this seems overstated, given that many XLs exceed the expected BS3 crosslinking distance (typically around 35 Å, with some reported exceeding 100 Å). This statement should be tempered and clarified, possibly acknowledging model limitations or alternative explanations.

Yes, there are indeed many long-distance crosslinks, particularly for the hetero-tetramer, and we agree with the reviewer that the original language mentioned was hyperbolic. The section has been rewritten, and is more conservative in tone in the revised manuscript. We do believe that the significant increase in the percentage of crosslinks that fall within the 35 Å cutoff for the hetero-octamer provides evidence that the hetero-tetramer is more dynamic, which we have several lines of evidence for. This is stated explicitly in the text:

“That a higher percent of the crosslinks comports with the cryo-EM model suggest the octamer is a less dynamic assembly, consistent with our biochemical studies (**Figure 1; Figure S1**).”

• Since “mass spectrometry also identified several contaminating β-barrel-containing proteins” that copurify with Augmin through multiple steps (as mentioned in the Supplementary list), these proteins should have been included in the database search used for the XL-MS analysis. Identifying binding sites between these β-barrel proteins and Augmin could have been helpful for structural interpretation. If insufficient XLs were obtained from co-purified complexes, the authors might consider performing a dedicated XL-MS experiment on the complex formed between the full Augmin hetero-octamer (AUG1–8) and the NEDD1-WD40 β-propeller domain.

We have performed two versions of the XL-MS analysis. One in which only the Augmin subunits are included in the database search, and one in which bacterial contaminant proteins are included (46 for hetero-tetramer and 42 hetero-octamer respectively, based on contaminants

with the highest spectral matches in the linear search (Figure S12)). In the searches that include only Augmin, we found 216 and 115 unique crosslinks respectively; in the searches with expanded databases these go down to 163 and 89.

On the balance, we consider the results from the more restricted search database more authoritative because they generated more identifications, and because the crosslinker we used (BS3) is non-cleavable and hence was not designed to be used for a larger database search. Hence, the files we deposit to PRIDE, the supplemental data, and the figures reflect this version of the search.

We do think that it is useful to include the list of potential co-purifying proteins and the crosslinking map including bacterial contaminants in the paper, and we have done so (in the form of Figure S12 and Data S1). We think these provide preliminary data to assign the additional density to RplB β -barrel ribosomal protein. But we do not think that they should be the basis for the “primary” structural dataset of the XL-MS studies.

Minor issues to be addressed:

Line 129: Figure 1A does not illustrate the presence or absence of conserved α -helices; it only shows the construct length and the position of the tags.

We have revised the text to correct this and removed the figure citation and added a reference citation.

Similarly, Line 139: Figure 1A does not provide any relevant information regarding the statement “Mass spectrometry confirms each Augmin subunit is purified in the assembly and that the AUG6 C-terminal region is prone to degradation with multiple polypeptide bands.” This should be clarified or corrected.

We revised the figure callout to be figure S1E for the statement above, which shows the percentile coverage of each of the subunits based on mass spec results.

Line 141: The manuscript references mass spectrometry data with “(Supplementary list file),” but the actual data are missing. Please ensure that this file is included.

We have provided Data S1 which enumerates all identified crosslinks

Line 348: It is unclear which complex (tetramer or octamer) the reported values “50% (105 out of 209)” and “70% (80 out of 115)” refer to. This should be explicitly stated.

We have revised the text to clarify which complex is which and address this

Dear editor and reviewers,

We are submitting a further revised version of our manuscript #NCOMMS-25-15494 entitled “Cryo-EM structures of the Plant Augmin reveal its intertwined coiled-coil assembly, antiparallel dimerization and NEDD1 binding mechanisms” We thank reviewers 1 and 3 for their attention to detail on the mass spectrometry data. We have made a few revisions to the text and to the supporting data file to rectify the matters that they have raised.

Please see a point-by-point response below.

Our responses are in blue, and the small changes to the text are bold-faced.

Reviewer #1 (Remarks to the Author):

Regarding my minor point #1, I continue to believe that the authors should revise the text in line 82. Additionally, identifying RplB as the globular density in the cryo-EM map is not very convincing. If RplB tightly binds to augmin, it should be possible to detect the RplB band on SDS-PAGE and easily identify it by traditional mass spectrometry along with augmin subunits (line140-141).

We agree that the assignment of the mystery data to RplB is at best, our best guess. It is supported by XL-MS and “traditional” mass spectrometry, as the reviewer suggests it ought to have been.

We have made a few modifications to the text to make it clearer that the assignment is tentative, and that RplB-derived tryptic peptides were indeed robustly identified as an abundant contaminant in the “traditional” linear peptide mass spec searches.

“The V-junction consistently possesses a high electron density spot. Mass spectrometry analysis **is suggestive that this interactor is RplB...**”

“A globular density, representing a bright spot residing inside the V-junction, **potentially corresponds to the ribosomal protein, RplB. Mass spectrometry on the Augmin pulldowns shows RplB to be an abundant contaminant with 518 peptide-spectrum matches, and moreover RplB crosslinks to AUG5 (see below; Figure S12). A β -barrel domain was placed in this density but was excluded from our model building and interpretation (Figure 1E, panel V; Figure S5B, G).**”

Reviewer #2 (Remarks to the Author):

The authors have addressed my questions. I have no further comments.

Reviewer #3 (Remarks to the Author):

I would like to thank the authors for their efforts in addressing my previous remarks and for improving their data analysis, particularly concerning the XL-MS part of their paper entitled “Cryo-EM structures of the Plant Augmin reveal its intertwined coiled-coil assembly, anGparallel dimerization, and NEDD1 binding mechanisms.” I also appreciate their willingness to make the raw data easily accessible, thereby ensuring transparency and open access to their research.

After this revision, the XL-MS section is now much clearer, easier to follow, and aligns better with the structural models proposed for the Augmin assemblies.

Nevertheless, before this work can be published—as it indeed deserves—there remain some discrepancies between the raw data and the data presented in the paper that must be resolved. Specifically, when examining the raw exports from the XiSearch search engine after FDR validation (i.e., links_xiFDR2.3.2.csv files), I could not find several cross-links (XLs) reported in the paper’s dataset (Dataset S1 Source file XLS). Neither the total number of XLs nor the specific XL amino acid pairs match.

Initially, I considered that redundancy might explain the differences; however, after confirming that there were no duplicates or decoys, discrepancies persisted. For example, in the tetrameric XL dataset, the paper reports 216 cross-links (consistent with the text), whereas the T8_15_combined_Links_xiFDR2.3.2.csv file in the PRIDE repository contains 277. Similarly, in the octameric XL experiment, 115 cross-links are reported in the paper but 172 appear in the raw export from PRIDE. If the authors applied additional filters beyond the FDR, these criteria should be explicitly stated and justified.

We thank the reviewer for paying close attention to this issue. The reason for the discrepancy was because Dataset S1 only reported on the crosslinks that could be mapped to residues that were included in the cryoEM reconstructed model. On the other hand, the outputs from xiFDR include all crosslinks that were confidently identified, regardless of whether they could be mapped to the structure or not.

To rectify the issue, Dataset S1 has been modified. Tabs 1 and 2 (which provide a list of all the unique crosslinks in AUG1,3,4,5 and AUG1,2,3,4,5,6,7,8 respectively) now provides a comprehensive list of crosslinks (277 and 172 respectively). It is obvious from the spreadsheet that the additional 61 and 57 crosslinks (at the bottom of the lists) are not in the model because they lack values in columns K, L, and M which assigns the crosslink a distance and PDB chain identifiers.

To make sure that this exclusion of some of crosslinks is clear to the reader, we have slightly amended the language in the text accordingly:

“Plotting the C α -C α distances for **the 216 crosslinks that can be fit within the Augmin hetero-tetramer (AUG1,3,4,5) cryo-EM model** reveals that 122 (56%) of these crosslinks fall below a 35 Å cut-off, and are shown **in Figure 5A (see also Figure S11A, D, G).**”

“Plotting the C α -C α distances for the 115 identified crosslinks that can be fit within the Augmin hetero-octamer (AUG1,2,3,4,5,6,7,8) cryo-EM model reveals that 85 (74%) of these crosslinks fall below a 35 Å cut-off, and are shown on the model in Figure 5B (see also Figure S11B, E, H).”

Moreover, despite some cross-links being absent from the raw export, others appear in the dataset presented in the paper but are not found in the original data. When comparing the octameric XL lists, I identified 26 cross-links that are present in the paper but absent from the raw export. While it is understandable that certain XLs might be excluded due to stricter filtering (which should also be explained), it is unclear how non-identified XLs could appear in the published dataset. The authors should correct this inconsistency.

After carefully reviewing the files, we see that the reviewer made the small mistake of not considering the possibility that the order of which crosslink site is called site1 and which crosslink site is called site2 can sometimes get permuted between xiFDR and xiView (the latter was used to generate the output file that became adapted into Dataset1).

To show that we have done our due diligence and can account for every single crosslink, we have supplied two supplemental spreadsheets for review purposes only (not to be published). They are called tetrameric_xlms and octameric_xlms.

In tetrameric_xlms, we have two tabs. The first tab, in columns A–M, is a reproduction of Dataset S1’s report for AUG1,3,4,5. In column N, we provide an identifier for each crosslink by concatenating protein1, residue1, protein2, residue 2, and in column O, we provide the alternative identifier for that crosslink if site1 and site2 were interchanged. The second tab provides the xiFDR output for AUG1,3,4,5 in columns A-AP, and in columns AQ and AR we provide identifiers for each crosslink using the same format, again allowing for interchange. In column P of the first tab, we use an Excel formula that asks whether this crosslink in Dataset S1 is also in the xiFDR report (in either the same or altered order). All 277 are found in the xiFDR report. In column AS of the second tab, we use the same Excel formula to ask if this crosslink in the xiFDR report is also in Dataset S1. All 277 are found in Dataset S1.

The file called octameric_xlms does the same quality control check for AUG1,2,3,4,5,6,7,8 and also shows 100% consistency.

Minor Comments

Lines 391–394:

“Because the dimer model lacks the V-junction, only 59 of the 115 cross-links could be mapped to the dimer model, but of those 59, 32 had even shorter C α –C α distances in the anti-parallel dimer model, suggesting the presence of Augmin assemblies in the anti-parallel dimer state in solution (Figure 5C; Figure S11C,F).”

This sentence is overly result-oriented. The fact that a cross-link exhibits a shorter C α -C α distance in one model compared to another does not necessarily imply stronger support for that model, especially when both fall within the range of a cross-linker. The probability of forming a cross-link (crosslinkability) is determined by multiple structural and dynamic factors. For a discussion of such criteria, please refer to: Filella-Merce, I. et al. (2020). "Quantitative Structural Interpretation of Protein Crosslinks."

While we agree that many factors influence the propensity for two residues to form a crosslink (with distance only being one), we do maintain that this comparison is a reasonable one because it compares distances for the same residue pair. Hence, other factors that contribute to crosslinkability (like residue chemistry, solvent exposure) for a given residue pair is nearly identical between the two structural models (monomer/dimer) we're comparing.

That said, we agree with the reviewer in spirit, and so have softened the language used in the text. Now we say that including the anti-parallel dimer state allows to better capture some of the crosslinking data instead of asserting that it directly implies that the anti-parallel dimer is physically present:

"Because the dimer model lacks the V-junction, only 59 of the 115 crosslinks could be mapped to the dimer model, but of those 59, 32 had even shorter C α -C α distances in the anti-parallel dimer model, **suggesting that inclusion of the anti-parallel dimer state allows us to better capture some of the crosslinking data.**"